



# Retrieving H$_2$O/HDO columns over cloudy and clear-sky scenes from the Tropospheric Monitoring Instrument (TROPOMI)

Andreas Schneider[1,a], Tobias Borsdorff[1], Joost aan de Brugh[1], Alba Lorente[1], Franziska Aemisegger[2], David Noone[3], Dean Henze[4], Rigel Kivi[5], and Jochen Landgraf[1]

[1]Earth science group, SRON Netherlands Institute for Space Research, Utrecht, the Netherlands
[2]Atmospheric Dynamics group, Department of Environmental Systems Science, ETH Zürich, Zürich, Switzerland
[3]Department of Physics, University of Auckland, Auckland, New Zealand
[4]Department of Ocean, Earth and Atmospheric Sciences, Oregon State University, Corvallis, Oregon, United States of America
[5]Earth Observation Research Unit, Finnish Meteorological Institute, Sodankylä, Finland
[a]Now at: Earth Observation Research Unit, Finnish Meteorological Institute, Sodankylä, Finland

**Correspondence:** Andreas Schneider (andreas.schneider@fmi.fi), Tobias Borsdorff (t.borsdorff@sron.nl)

**Abstract.** This paper presents an extension of the scientific HDO/H$_2$O column data product from the Tropospheric Monitoring Instrument (TROPOMI) including clear-sky and cloudy scenes. The retrieval employs a forward model which accounts for scattering, and the algorithm infers the trace gas column information, surface properties and effective cloud parameters from the observations. The extension to cloudy scenes greatly enhances coverage, particularly enabling data over oceans. The data

set is validated against co-located ground-based Fourier transform infrared (FTIR) observations by the Total Carbon Column Observing Network (TCCON). The median bias for clear-sky scenes is $1.4 \times 10^{21}$ molec cm$^{-2}$ (2.9 %) in H$_2$O columns and $1.1 \times 10^{17}$ molec cm$^{-2}$ (−0.3 %) in HDO columns, which corresponds to −17 ‰ (9.9 %) in a posteriori $\delta$D. The bias for cloudy scenes is $4.9 \times 10^{21}$ molec cm$^{-2}$ (11 %) in H$_2$O, $1.1 \times 10^{17}$ molec cm$^{-2}$ (7.9 %) in HDO, and −20 ‰ (9.7 %) in a posteriori $\delta$D. At low-altitude stations in low and middle latitudes the bias is small, and has a larger value at high latitude stations. At

high altitude stations, an altitude correction is required to compensate for different partial columns seen by the station and the satellite. The bias in a posteriori $\delta$D after altitude correction depends on sensitivity due to shielding by clouds, and on realistic prior profile shapes for both isotopologues. Cloudy scenes generally involve low sensitivity below the clouds, and since the information is filled up by the prior, it plays an important role in these cases. Over oceans, aircraft measurements with the Water Isotope System for Precipitation and Entrainment Research (WISPER) instrument from a field campaign in 2018 are

used for validation, yielding a bias of −3.9 % in H$_2$O and −3 ‰ in $\delta$D over clouds. To demonstrate the added value of the new data set, a short case study of a cold air outbreak over the Atlantic Ocean in January 2020 is presented, showing the daily evolution of the event with single overpass results.

## 1 Introduction

Atmospheric moisture strongly controls Earth's radiative budget and transports energy via latent heat, e.g. from low to high

latitudes. Uncertainties in the quantification of these two effects are still large and represent one of the key uncertainties in



current climate prediction (Stevens and Bony, 2013). Isotopologues of water offer further insights into the water cycle due to fractionation processes on phase changes. This provides additional constraints for models and thus valuable insights for their improvement. The application of isotopic effects to this end requires observations on global scale and with a long-term perspective, whereto satellite observations from space are most promising (Rast et al., 2014).

HDO and $H_2O$ are observed from space mainly in the thermal infrared spectral range, e. g. by the Infrared Atmospheric Sounding Interferometer (IASI) onboard the MetOP satellites (Herbin et al., 2009; Schneider and Hase, 2011; Schneider et al., 2016; Lacour et al., 2012) or the Atmospheric Infrared Sounder (AIRS) onboard the NASA Aqua satellite (Worden et al., 2019) which builds on earlier work using the Tropospheric Emission Spectrometer (TES) on the NASA Aura satellite (Worden et al., 2012). These sounders can observe clear-sky and cloudy scenes over land and oceans, but they are insensitive to the boundary

layer. The short-wave infrared (SWIR) spectral range does provide sensitivity to the boundary layer and is suitable to estimate total columns, however bodies of water are very dark in the SWIR which makes retrievals over oceans impossible for clear-sky conditions. The Tropospheric Monitoring Instrument (TROPOMI) onboard the Sentinel 5 Precursor (S5P) satellite launched on 13 October 2017 (Veefkind et al., 2012) will, together with its successor instrument Sentinel 5 on MetOp-SG-A, provide measurements in the SWIR beyond the year 2040 with unprecedented spatial resolution of $7\,\text{km} \times 7\,\text{km}$ (upgraded to $5.5\,\text{km} \times$

$7\,\text{km}$ in August 2019) in the centre of the swath, daily global coverage and superior radiometric performance. Schneider et al. (2020) have recently published a first clear-sky data set of $H_2O$ and HDO columns from TROPOMI. However, the restriction to clear-sky scenes over land hinders hydrological studies: cloudy-sky conditions are often different from clear-sky conditions, and oceans are important for the hydrological cycle. In order to extend the coverage to cloudy scenes, and therewith also to oceans, an updated retrieval is employed which accounts for scattering and estimates effective cloud parameters additionally

to the trace gases. Any loss of sensitivity to the partial column below the cloud is notified by the retrieval algorithm to improve data interpretation.

Isotopological abundance variations are often described by the so-called $\delta$ notation which denotes the relative difference of the ratio of the heavy and the light isotopologue, $R_{\text{HDO}} = c_{\text{HDO}}/c_{\text{H}_2\text{O}}$, to the standard abundance ratio of Vienna Standard Mean Ocean Water (VSMOW) $R_{\text{HDO,std}} = 3.1152 \times 10^{-4}$, i. e.

$$\delta\text{D} = \frac{R_{\text{HDO}} - R_{\text{HDO,std}}}{R_{\text{HDO,std}}} \tag{1}$$

(Craig, 1961b; Hagemann et al., 1970). This nomenclature is also used herein.

Section 2 describes the retrieval setup, detailing the changes compared to the previous clear-sky-only data product by Schneider et al. (2020). Section 3 introduces reference data used for validation and intercomparison, namely ground-based Fourier transform infrared (FTIR) observations over land and aircraft measurements over the ocean. Section 4 shows validation re-

sults, with low-altitude and high-altitude FTIR stations presented separately. A comparison to the clear-sky data product by Schneider et al. (2020) for the same ground pixels is also included. Over the ocean, the retrievals are compared to aircraft measurements. Section 5 presents applications of the new data set on the global scale as well as locally for single overpasses. Finally, Sec. 6 gives a summary and conclusions.



## 2 Retrieval method

This work employs the Shortwave Infrared CO Retrieval (SICOR) algorithm, which utilises a profile-scaling approach; it is described in detail by Scheepmaker et al. (2016), Landgraf et al. (2016) and Borsdorff et al. (2014). While the clear-sky retrieval by Schneider et al. (2020) employs a forward model which ignores scattering (hereafter non-scattering retrieval), the update presented herein uses a forward model which does account for scattering using the Practical Improved Flux Method (PIFM, Zdunkowski et al., 1980) and is termed scattering retrieval hereafter. The target trace gases $H_2O$ and HDO are fitted together

with the interfering species $CH_4$ and CO and a Lambertian surface albedo in the spectral window from 2354.0 nm to 2380.5 nm (Scheepmaker et al., 2016). The isotopologue $H_2^{18}O$ is included in the forward model but not fitted. Scattering cross-sections are taken from the high-resolution transmission molecular absorption database (HITRAN) 2016 release (Gordon et al., 2017). A priori profiles of water vapour are taken from the European Centre for Medium-Range Weather Forecasts (ECMWF) analysis product. Since the ECMWF data product does not distinguish individual isotopologues, $H_2O$, HDO and $H_2^{18}O$ profiles are

obtained from the water vapour profile by scaling it with the respective average relative natural abundances. That implicitly corresponds to a prior of $\delta D$ of 0‰. A case study for high-altitude stations in Sec. 4.2 alternatively uses HDO prior profiles computed from $H_2O$ profiles via an assumed more realistic $\delta D$ profile which linearly decreases from $-100$‰ at the surface to $-600$‰ at 15 km altitude followed by a linear increase to $-400$‰ at the top of the atmosphere as used by Scheepmaker et al. (2016) for their simulated measurements. From this $\delta D$ profile, a $\delta^{18}O$ profile is computed via the global meteoric water line

$$\delta D = 8\,\delta^{18}O + 10‰ \tag{2}$$

(Craig, 1961a) and used to obtain the $H_2^{18}O$ prior profile from the $H_2O$ profile. A priori profiles of $CH_4$ and CO are taken from TM5 simulations (Krol et al., 2005).

Clouds are modelled by a single scattering layer with a triangular height profile in extinction coefficient centred at cloud centre height $h$ with a geometrical half-width $d$ and a cloud optical thickness of $\tau$, deploying a two-stream model. The idea is

to infer these effective cloud parameters from deviations of the retrieved methane column to the prior, as such differences are supposed to originate from light path modifications by scatterers. Fitting both $d$ and $\tau$ would lead to ambiguities, thus the cloud geometric thickness $d$ is fixed at 2500 m. The sensitivity of the inferred cloud parameters on the actual choice of $d$ is relatively small. The approach of the CO product (Landgraf et al., 2016), which comprises fitting $h$ and $\tau$ simultaneous to the trace gases in its spectral range 2315–2338 nm, cannot directly be transferred to the spectral window 2354–2380.5 nm because it introduces

errors in the inferred water vapour columns, maybe due to interferences and/or inaccuracies of the methane spectroscopy in the latter window. Thus, the effective cloud parameters are determined in a pre-fit in the spectral window from 2310 nm to 2338 nm where large absorption features of methane not interfering with water vapour are present. The resulting parameters are taken over to the final fit in the spectral window from 2354.0 nm to 2380.5 nm, where they are fixed while the trace gases are fitted. This neglects the spectral dependence of the cloud optical thickness in the spectral range between 2310 nm and 2380 nm.

Figure 1 visualises the spectral windows employed for the retrieval in plots of simulated transmission spectra of the relevant absorbers.





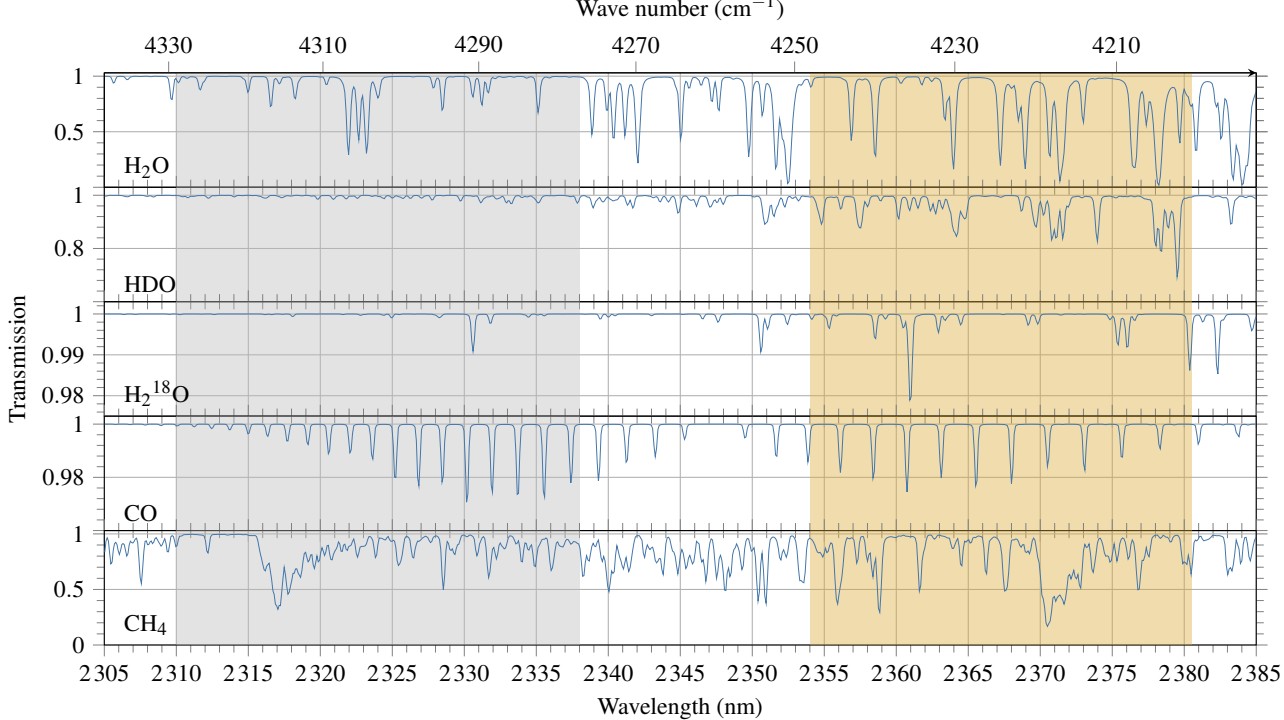

**Figure 1.** Simulation of atmospheric transmission in the spectral range of TROPOMI's SWIR channel for the absorbers taken into account by the retrieval algorithm. The grey shading marks the spectral window used for the determination of effective cloud parameters, the yellow shading the spectral window for the retrieval of the trace gases.

A priori surface albedos are taken from a one-year average over the year 2018 of the non-scattering product on an equal-area grid with $5760 \times 2880$ bins (corresponding to a resolution of $0.125°$ at the equator). Values over oceans and lakes (where the non-scattering retrieval does not yield data) are set to 0 as water is very dark in the short-wave infrared. Figure 2 shows a map

of this prior. To reduce interferences with cloud parameters and stabilise the inversion, the surface albedo is slightly regularised to the prior. Regularisation in the context of ill-posed problems is discussed in detail by Borsdorff et al. (2014).

The results are filtered for convergence and with a quality filter based on fit quality in terms of the number of iterations and $\chi^2$ as measure for the residual. Moreover, scenes with high solar zenith angles larger than $70°$ are filtered out since they are prone to errors. These errors are on one hand due to multi-scattering and diffraction effects not covered well by the two-stream

forward model, and on the other hand due to typically low radiances resulting in low signal-to-noise ratios. From the remaining data, scenes are classified as clear-sky, cloudy with low clouds, or other (e. g. high clouds) based on retrieved effective cloud parameters as specified in Tab. 1. Only scenes of the first two categories (i. e. clear-sky or low clouds) are considered in this study and recommended to be taken into account by the user, except if the data are assimilated using averaging kernels. Clear-sky scenes are additionally filtered for surface albedo because low surface albedos usually involve low signal-to-noise ratios.



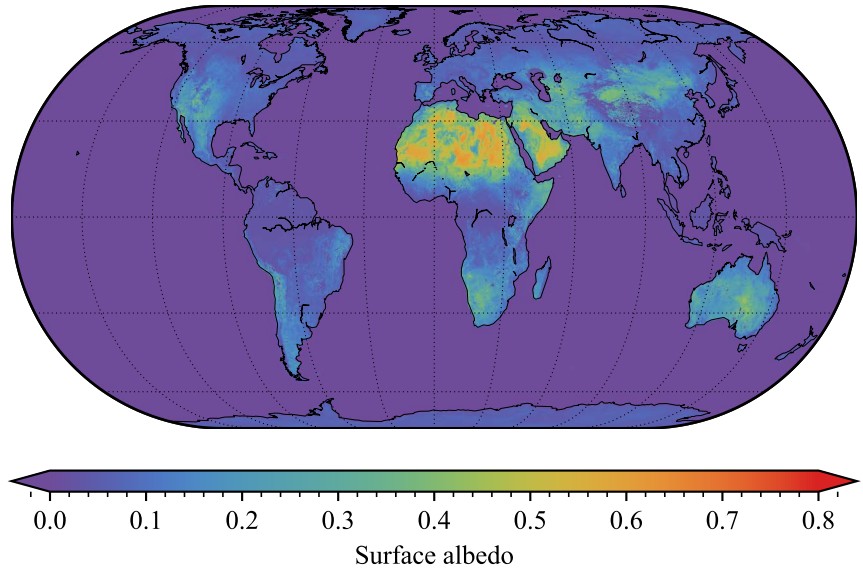

**Figure 2.** Average surface albedo from the non-scattering retrieval (Schneider et al., 2020) of the year 2018, which is used as a priori surface albedo for the scattering retrieval. Values over oceans and lakes (where the non-scattering retrieval does not yield data) are set to 0.

**Table 1.** Quality filters and selection criteria for clear-sky and cloudy-sky conditions.

| Quantity | Filter |
|---|---|
| Quality filter (all scenes) | |
| Number of iterations | $n \leq 10$ |
| Reduced $\chi^2$ | $\chi^2_{\mathrm{f}} \leq 150$ |
| Reduced $\chi^2$ of pre-fit | $\chi^2_{\mathrm{p}} \leq 150$ |
| Solar zenith angle | $\vartheta \leq 70°$ |
| Clear-sky filter | |
| Cloud optical thickness | $\tau_{\mathrm{cld}} < 0.3$ |
| Surface albedo | $a \geq 0.02$ |
| Filter for cloudy scenes | |
| Cloud height | $h_{\mathrm{cld}} \leq 2000\,\mathrm{m}$ |
| Cloud optical thickness | $\tau_{\mathrm{cld}} > 0.3$ |





Such a surface albedo filter is not applied to cloudy scenes because clouds usually have high reflectivity and shield the surface, which allows to retrieve over very low surface albedos with high signal-to-noise ratio.

Retrievals over optically thick clouds are insensitive to the partial column below the cloud. The algorithm estimates the missing information from the prior, however that can deviate from the truth. This requires a thorough data interpretation using the column averaging kernel, which indicates the vertical retrieval sensitivity. It can be used, e. g., to assimilate the data with

models to help with the interpretation when sensitivity is low.

## 3   Reference data

### 3.1   Ground-based measurements by TCCON and co-location criteria

To validate the new satellite data set, ground-based Fourier transform infrared (FTIR) observations by the Total Carbon Column Observing Network (TCCON, Wunch et al., 2011), version GGG2014 are used. The TCCON HDO data are bias-corrected by

dividing the HDO columns by a correction factor of 1.0778 as derived by Schneider et al. (2020). This correction accounts for a missing calibration. Table 2 lists the stations that are used for the validation.

An FTIR instrument has sensitivity in its viewing direction (i. e. in direction of the sun). If the sun is low in the sky (i. e. for high solar zenith angles), this translates into an azimuthal dependency of sensitivity, while there is no azimuthal dependency if the sun is in the zenith. To take this into account, the spatial co-location considers satellite overpasses in a cone in FTIR

viewing direction with an opening angle $\alpha$ and a radius $r_\alpha$ depending on solar zenith angle $\vartheta$. Varying the opening angle linearly with SZA from $\alpha_0$ at $\vartheta = 90°$ to $360°$ at $\vartheta = 0°$ and requiring equal co-location area in all cases gives

$$\alpha(\vartheta) = \alpha_{90} + \frac{90° - \vartheta}{90°}(360° - \alpha_{90}) \tag{3}$$

$$r_\alpha = \sqrt{\frac{360°}{\alpha}} \, r_0. \tag{4}$$

Figure 3 illustrates this condition, which selects ground pixels depending on the directional sensitivity of the FTIR while

keeping the co-location area constant. Here, $\alpha_{90} = 45°$ is selected and $r_0$ is computed from the radius at a solar zenith angle of $90°$, $r_0 = \sqrt{\frac{\alpha_{90}}{360°}} \, r_{90}$ with $r_{90°} = 30\,\text{km}$. With these selections, the limit of $\vartheta = 90°$ gives the co-location criteria used for the validation of the non-scattering retrieval by Schneider et al. (2020).

Additionally, the time between satellite and ground measurements has to be less than 2 h to minimise representation errors due to the diurnal cycle. Since the FTIR has to directly see the sun (possibly through gaps in the clouds) to take measurements,

co-located cloudy satellite observations require a change in the cloud cover within the co-location radius or the co-location time.

At low-altitude stations (i. e. stations below 1000 m above mean sea-level (a. s. l.), only TROPOMI ground pixels with an altitude difference to the station height of less than 500 m are used. If the altitude difference is too large, the observation of different partial columns leads to errors. High-altitude stations are treated separately in Sec. 4.2.





**Table 2.** List of TCCON ground stations used for the validation.

| Station | Latitude | Longitude | Altitude | Data available from/to | Reference |
|---|---|---|---|---|---|
| Eureka | 80.1° N | 86.4° W | 610 m | 24 Jul 2010 – 07 Jul 2020 | Strong et al. (2019) |
| Ny Ålesund | 78.9° N | 11.9° E | 20 m | 28 Mar 2006 – 06 Jul 2019 | Notholt et al. (2019b) |
| Sodankylä | 67.4° N | 26.6° E | 190 m | 16 May 2009 – 30 Oct 2019 | Kivi et al. (2014) |
| East Trout Lake | 54.4° N | 105.0° W | 500 m | 07 Oct 2016 – 04 Jul 2020 | Wunch et al. (2018) |
| Bialystok | 53.2° N | 23.0° E | 190 m | 01 Mar 2009 – 01 Oct 2018 | Deutscher et al. (2019) |
| Bremen | 53.1° N | 8.9° E | 30 m | 15 Jan 2007 – 23 Aug 2019 | Notholt et al. (2019a) |
| Karlsruhe | 49.1° N | 8.4° E | 110 m | 19 Apr 2010 – 31 Jul 2020 | Hase et al. (2015) |
| Paris | 48.8° N | 2.4° E | 60 m | 23 Sep 2014 – 23 Jul 2019 | Té et al. (2014) |
| Orléans | 48.0° N | 2.1° E | 130 m | 29 Aug 2009 – 31 Jul 2019 | Warneke et al. (2019) |
| Garmisch | 47.5° N | 11.1° E | 750 m | 16 Jul 2007 – 18 Oct 2019 | Sussmann and Rettinger (2018a) |
| Zugspitze | 47.4° N | 11.0° E | 2960 m | 24 Apr 2015 – 17 Oct 2019 | Sussmann and Rettinger (2018b) |
| Park Falls | 45.9° N | 90.3° W | 440 m | 02 Jun 2004 – 02 Apr 2020 | Wennberg et al. (2017) |
| Rikubetsu | 43.5° N | 143.8° E | 380 m | 16 Nov 2013 – 31 Jul 2019 | Morino et al. (2018c) |
| Lamont | 36.6° N | 97.5° W | 320 m | 06 Jul 2008 – 01 Apr 2020 | Wennberg et al. (2016b) |
| Tsukuba | 36.0° N | 140.1° E | 30 m | 04 Aug 2011 – 31 Jul 2019 | Morino et al. (2018a) |
| Edwards | 35.0° N | 117.9° W | 700 m | 20 Jul 2013 – 04 Jul 2020 | Iraci et al. (2016) |
| JPL | 34.2° N | 118.2° W | 390 m | 19 May 2011 – 14 May 2018 | Wennberg et al. (2016a) |
| Pasadena | 34.1° N | 118.1° W | 240 m | 20 Sep 2012 – 03 Jul 2020 | Wennberg et al. (2015) |
| Saga | 33.2° N | 130.3° E | 10 m | 28 Jul 2011 – 04 May 2020 | Kawakami et al. (2014) |
| Izaña | 28.3° N | 16.5° W | 2370 m | 18 May 2007 – 31 Jul 2020 | Blumenstock et al. (2017) |
| Burgos | 18.5° N | 120.7° E | 40 m | 03 Mar 2017 – 22 Aug 2019 | Morino et al. (2018b) |
| Wollongong | 34.4° S | 150.9° E | 30 m | 25 Jun 2008 – 31 Jul 2019 | Griffith et al. (2014) |
| Lauder | 45.0° S | 169.7° E | 370 m | 02 Feb 2010 – 04 May 2020 | Sherlock et al. (2014), Pollard et al. (2019) |

The effects by different a priori profiles used by FTIR and satellite retrievals are accounted for with the column averaging kernel. Following Borsdorff et al. (2014), the adjustment of column $c_i$ retrieved using prior profile $\mathbf{x}_{ai}$ to prior profile $\mathbf{x}_{aj}$ is performed with the column averaging kernel $\mathbf{A}_i$ of retrieval $i$ by

$$c_s = c_i + (\mathbf{1} - \mathbf{A}_i)^T \mathbf{x}_{aj} \tag{5}$$

where $\mathbf{1}$ is a vector of ones. In the present case, $i$ is TROPOMI and $j$ is TCCON. TCCON prior profiles are linearly interpo-
lated from TCCON levels to SICOR layer centres, and the top layer is extended to 0 Pa. This correction is performed for all comparisons with TCCON data except for high-altitude stations.



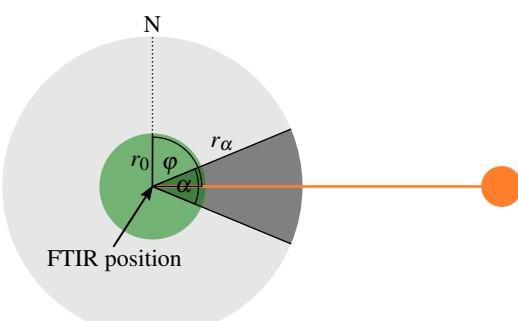

**Figure 3.** Illustration of the spatial co-location condition. The co-location area consists of a cone in FTIR viewing direction (i. e. solar azimuth angle $\varphi$) with opening angle $\alpha$ and radius $r_\alpha$ depending on solar zenith angle $\vartheta$ (dark grey). The limit of $\vartheta = 0°$ is a full circle (green). The area remains constant (dark grey and green).

## 3.2 Ground-based measurements by MUSICA-NDACC

The project MUlti-platform remote Sensing of Isotopologues for investigating the Cycle of Atmospheric water (MUSICA, Schneider et al., 2016; Barthlott et al., 2017) also provides a ground-based water vapour isotopologue data product, which
uses spectra measured within the Network for the Detection of Atmospheric Composition Change (NDACC, De Mazière et al., 2018). Two different products exist: firstly the direct retrieval output, called type 1 product, and secondly an a posteriori processed output that reports the optimal estimation of ($H_2O$, $\delta D$) pairs, called type 2 product. Here, the type 2 product is used because it is recommended for isotopologue analyses (Barthlott et al., 2017). Recent MUSICA-NDACC data are currently only available for three stations (Karlsruhe, Kiruna and Izaña), which compromises globally valid validation studies.
Seven stations are in both networks TCCON and NDACC. In theses cases, the TCCON and NDACC measurements are performed with the same instrument, but in a different spectral range at different times. As shown e. g. by Schneider et al. (2020), the retrievals from the two networks are biased to each other.

**Table 3.** List of ground stations used for the derivation of the FTIR correction.

| Station | Lat. | Lon. | Altitude | MUSICA available from/to | TCCON available from/to | TCCON reference |
|---|---|---|---|---|---|---|
| Eureka | 80° N | 86° W | 610 m | 01 Aug 2006 – 01 Sep 2014 | 24 Jul 2010 – 07 Jul 2020 | Strong et al. (2019) |
| Ny Ålesund | 79° N | 12° E | 20 m | 08 Apr 2005 – 27 Aug 2014 | 28 Mar 2006 – 14 May 2018 | Notholt et al. (2019b) |
| Bremen | 53° N | 9° E | 30 m | 21 Jul 2004 – 14 Oct 2014 | 15 Jan 2007 – 29 May 2018 | Notholt et al. (2019a) |
| Karlsruhe | 49° N | 8° E | 110 m | 17 Apr 2010 – 12 Sep 2019 | 19 Apr 2010 – 25 Jun 2020 | Hase et al. (2015) |
| Izaña | 28° N | 17° W | 2370 m | 18 Jun 2001 – 25 Sep 2019 | 18 May 2007 – 30 Jun 2020 | Blumenstock et al. (2017) |
| Wollongong | 34° S | 151° E | 30 m | 07 Aug 2007 – 09 Sep 2014 | 25 Jun 2008 – 31 Jul 2019 | Griffith et al. (2014) |
| Lauder | 45° S | 170° E | 370 m | 06 Sep 1997 – 30 Aug 2014 | 02 Feb 2010 – 31 Oct 2018 | Sherlock et al. (2014) |

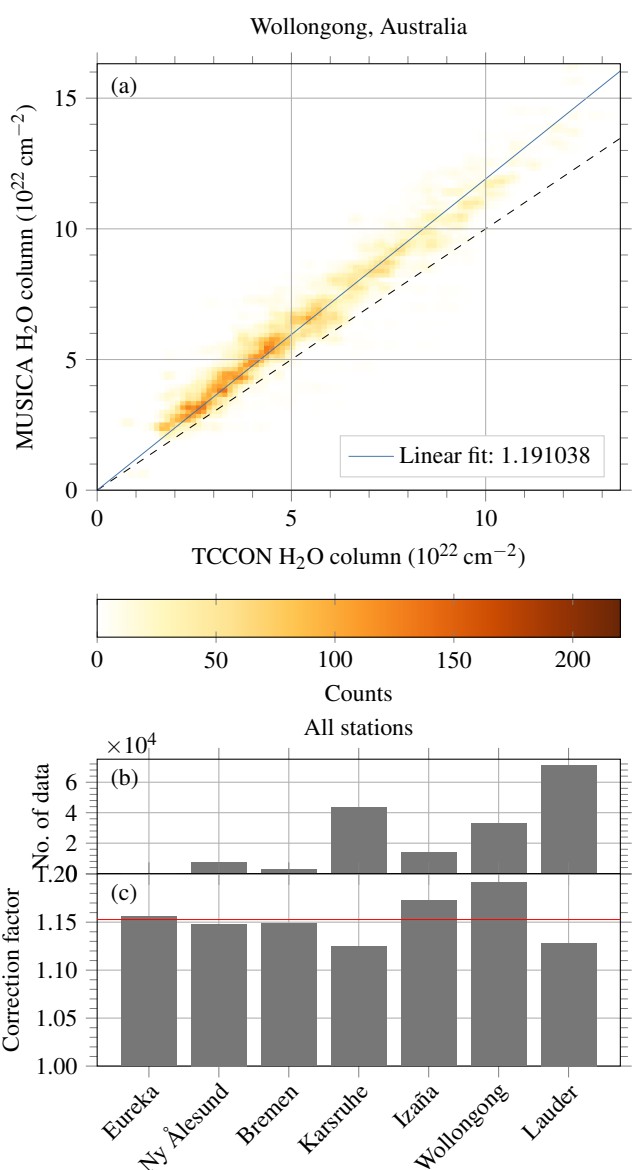

**Figure 4. (a)** Histogram of co-located TCCON and MUSICA-NDACC $H_2O$ columns at Wollongong (colour-coded) and result of a fit of a linear correction (blue line). **(b)** Number of co-located observations for all individual stations in both networks. **(c)** Correction factors to correct MUSICA-NDACC $H_2O$ columns to TCCON. The average 1.1527 is marked by a red line.



Based on the fact that MUSICA $\delta$D is calibrated by aircraft measurements near Izaña but TCCON HDO is not verified, Schneider et al. (2020) derived a correction of TCCON HDO by matching TCCON a posteriori $\delta$D to MUSICA-NDACC $\delta$D.

Nevertheless, also $H_2O$ columns differ between TCCON and MUSICA-NDACC. Since TCCON $H_2O$ is better validated and thus assumed to be correct, this discrepancy is solved by a correction of MUSICA-NDACC derived below. Figure 4a shows correlations of TCCON and MUSICA-NDACC $H_2O$ columns at Wollongong, Australia. The difference is well described by a simple scaling of the column. The result of such a fit for all stations in both networks (as listed in Tab. 3) is presented in Fig. 4c. The correction factors do not vary considerably between stations. To harmonise both data sets, MUSICA $H_2O$

and HDO columns are thus corrected by division by the mean correction factor 1.1525 (red line in Fig. 4c). This adjusts the MUSICA $H_2O$ columns while leaving MUSICA $\delta$D unchanged.

Filling the null-space of TROPOMI measurements with MUSICA-NDACC prior profiles with averaging kernels creates large scatter and deviations from the reference. MUSICA a priori profiles do not depend on time and are much less realistic than TCCON or TROPOMI prior profiles. This can lead to deviations. Thus, averaging kernels are not applied for the validation

with MUSICA-NDACC data.

### 3.3 Aircraft measurements by the WISPER instrument

During the NASA ObseRvations of Aerosols above CLouds and their intEractionS (ORACLES) field mission in the southeastern Atlantic Ocean region (Redemann et al., 2021), measurements of $H_2O$ mixing ratio and $\delta$D were taken onboard the NASA P-3B Orion aircraft with the Water Isotope System for Precipitation and Entrainment Research (WISPER) instrument

(Henze et al., 2021). This instrument employs in situ gas phase cavity ring-down water vapour isotopic analysers (Picarro model L2120-fi) coupled to inlets that enable paired measurements of cloud water and total water amounts and isotope ratios..

The validation uses profile measurement data from the 2018 deployment. Only profiles reaching at least 5000 m are taken into account. For ascent profiles, descending sections are filtered out by discarding sections with higher pressure than a previous data point; similarly, ascending sections are removed from descent profiles. If more than 30 % of the data are discarded in this

step, the whole profile is dropped. This eliminates flight sections with a "saw-tooth" pattern designed for sampling in cloudy regions. Altogether, 17 profiles pass the filter, spanning the time range from 27 Sep 2018 to 21 Oct 2018. The top altitude varies between 5130 m and 7408 m with an average of 6195 m. The vertical resolution is typically 30 m due to sampling at 1 Hz and typical aircraft decent rates. HDO mixing ratios are computed from $H_2O$ mixing ratios and $\delta$D. In order to compute a total column corresponding to the profile, each measured mixing ratio is assumed to be constant until the data point below,

with the lowest measurement assumed constant down to the surface with surface pressure taken from the ECMWF analysis data product interpolated to the location of the profile.

The co-location is performed with the full 360° viewing angle (as the in situ instrument does not have a directional sensitivity like the FTIR) and a radius of 10.6066 km (corresponding to the radius for the full circle $r_0$ in Sec. 3.1). For each co-located measurement, the satellite prior profile is scaled such that the partial column below the ceiling of the aircraft profile coincides

that of the aircraft measurement. The aircraft profile is interpolated to the grid of the prior profile, and the part above the ceiling is complemented by the upper part of the scaled prior profile. Finally, the averaging kernel $\mathbf{A}_i$ of the satellite measurement is





applied to compute the smoothed reference column by

$$x_{\text{ref,s}} = \mathbf{A}_i^T \mathbf{x}_{\text{ref}}. \tag{6}$$

## 4  Validation

In the following subsections, the scattering retrieval is validated for clear-sky and cloudy scenes according to retrieved effective cloud parameters as described above in Sec 2. As reference, the plots additionally show the non-scattering retrieval filtered as reported by Schneider et al. (2020), i. e. with the cloud fraction from the Visible Infrared Imaging Radiometer Suite (VIIRS) co-located to the TROPOMI field of view, a two-band filter as described in loc. cit., and by solar zenith angle.

### 4.1  Low-altitude stations

Figure 5 depicts an exemplary time series of daily medians of co-located measurements at the TCCON station Karlsruhe. The TROPOMI observations follow the reference well, although some deviations are present especially for cloudy scenes. Figure 6 presents corresponding correlations. Retrieved columns correlate excellently to the reference with a Pearson coefficient of 0.98 in $H_2O$ and 0.99 in HDO for clear-sky scenes, and 0.95 in $H_2O$ and 0.96 in HDO for cloudy scenes. A posteriori $\delta D$ has more scatter with correlation coefficients of 0.86 and 0.83 for clear-sky and cloudy scenes, respectively. The bias, which is defined

as the mean difference between TROPOMI and TCCON, is for clear-sky scenes $-1.3 \times 10^{20}$ molec cm$^{-2}$ ($-0.4\,\%$) in $H_2O$ and $-3.6 \times 10^{16}$ molec cm$^{-2}$ ($-1.0\,\%$) in HDO, which corresponds to a bias in a posteriori $\delta D$ of $-3\,\%_o$ ($1.1\,\%$). For cloudy scenes, it is $4.9 \times 10^{21}$ molec cm$^{-2}$ ($8.3\,\%$) in $H_2O$, $1.1 \times 10^{18}$ molec cm$^{-2}$ ($6.5\,\%$) in HDO and $-12\,\%_o$ ($7.3\,\%$) in a posteriori $\delta D$. The retrieval performance for cloudy scenes is good: correlations are similar as for clear-sky scenes or the non-scattering retrieval, although the bias is larger. This can be explained by small sensitivity of the retrieval below optically thick clouds.

Figure 7 presents statistics and correlation coefficients of daily medians at all low-altitude stations. The amount of data for clear-sky scenes of the new scattering retrieval is much larger than for the old non-scattering retrieval: on average a factor of 8 more. This is connected to different filtering: while the non-scattering product is strictly filtered with the S5P-VIIRS product and an additional two-band filter (Schneider et al., 2020), the scattering product is filtered with effective cloud parameters retrieved in the pre-fit (see Table 1). The number of observations (ground pixels) per day (Fig. 7b) is usually around 4 but

significantly higher at high latitudes due to multiple overpasses per day. Cloudy scenes encounter typically less observations per day compared to clear-sky scenes with a median of 3.4 vs. 4.1. The non-scattering retrieval has a significantly lower data yield with a median of 2.7 co-located ground pixels per day. The distributions visualised by the violin plots show that there is quite some spread with some days with a high number of observations.

Correlations of daily medians of $H_2O$ and HDO columns are excellent at all stations (Fig. 7c, d). In a posteriori $\delta D$, cor-

relations are lower at some stations, typically ones with low seasonal variation (Fig. 7e). For clear-sky scenes, correlation coefficients are similar to those of the non-scattering product except for $\delta D$ at some stations like JPL and Pasadena. For cloudy scenes, the correlations are mostly slightly lower than for clear-sky scenes.



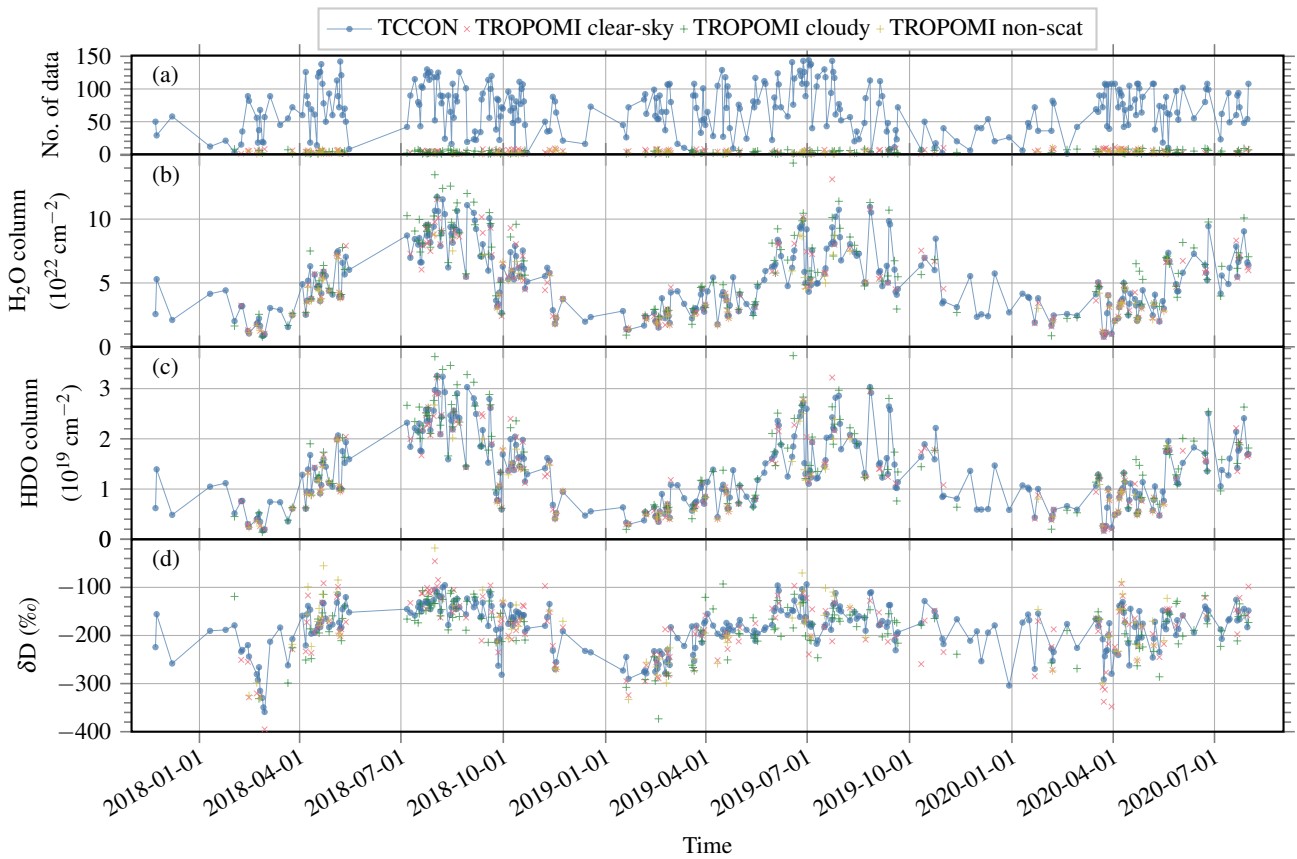

**Figure 5.** Time series of individual observations per day **(a)**, daily medians of $H_2O$ columns **(b)**, HDO columns **(c)** and a posteriori $\delta D$ **(d)** of TCCON (blue), TROPOMI clear-sky scenes (red), TROPOMI cloudy scenes (green), and the TROPOMI non-scattering retrieval (orange) at Karlsruhe, Germany (49.1° N, 8.4° E, 110 m a. s. l.)

Biases are depicted in Fig. 8. At low and middle latitudes the bias is generally small: at these stations, the median for clear-sky scenes is $1.3 \times 10^{21}$ molec cm$^{-2}$ (1.8 %) in $H_2O$ columns, $2.0 \times 10^{16}$ molec cm$^{-2}$ ($-0.3$ %) in HDO columns, and $-8$‰
(4.6 %) in $\delta D$, the one for cloudy scenes is $4.7 \times 10^{21}$ molec cm$^{-2}$ (8.8 %) in $H_2O$ columns, $1.1 \times 10^{18}$ molec cm$^{-2}$ (6.5 %) in HDO columns, and $-20$‰ (12 %) in $\delta D$. High-latitude stations mostly have larger biases that can be as high as 20 % in the columns and 40‰ in a posteriori $\delta D$. The median bias at high latitude stations (Eureka, Ny Ålesund, Sodankylä, and East Trout Lake) in $H_2O$, HDO and $\delta D$ is for clear-sky scenes $2.3 \times 10^{21}$ molec cm$^{-2}$ (9.5 %), $4.0 \times 10^{17}$ molec cm$^{-2}$ (0.4 %) and $-37$‰ (13 %) and for cloudy scenes $5.1 \times 10^{21}$ molec cm$^{-2}$ (12 %), $1.0 \times 10^{18}$ molec cm$^{-2}$ (9.1 %) and $-24$‰ (8.4 %),
respectively. These high biases are similar, but partly more pronounced than for the non-scattering retrieval. High-latitude locations employ difficult measurement geometries with typically high solar zenith angles and low surface albedos, in which the additional estimation of cloud parameters seems to be even more challenging. In summer, these biases are typically lower than in darker seasons with higher solar zenith angles. The bias is also high at Garmisch, which lies in a mountainous region



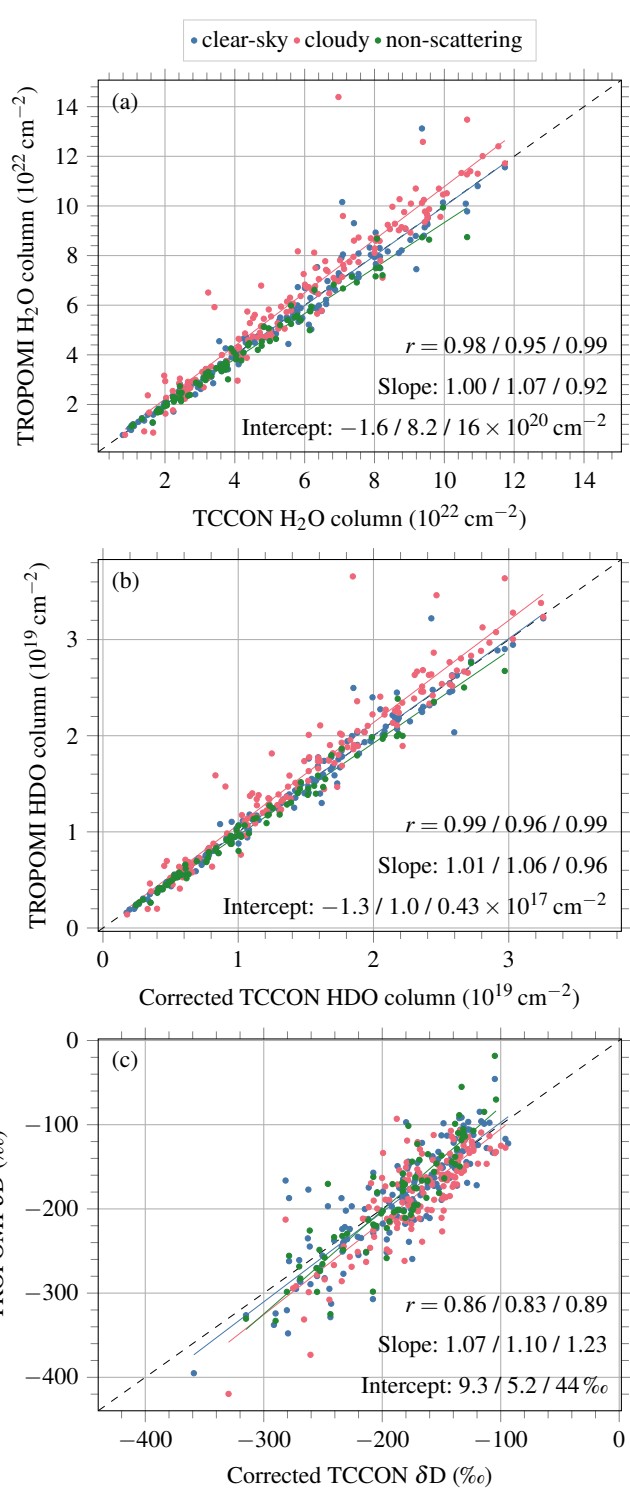

**Figure 6.** Correlations of TROPOMI observations against corrected TCCON measurements of $H_2O$ columns **(a)**, HDO columns **(b)** and a posteriori $\delta D$ **(c)** at Karlsruhe.

**Figure 7.** Number of days with observations **(a)**, observations per day **(b)**, correlation coefficients of $H_2O$ columns **(c)**, correlation coefficients of HDO columns **(d)**, and correlation coefficients of a posteriori $\delta D$ **(e)** at all TCCON stations.

meaning a typically complex topography with large variation in surface altitude and albedo within a ground pixel. The median

bias of all stations is for clear-sky scenes $1.4 \times 10^{21}$ molec cm$^{-2}$ (2.9 %) in $H_2O$ columns, $1.1 \times 10^{17}$ molec cm$^{-2}$ (−0.3 %) in HDO columns, and −17‰ (9.9 %) in a posteriori $\delta D$. For cloudy scenes, it is $4.9 \times 10^{21}$ molec cm$^{-2}$ (11 %) in $H_2O$, $1.1 \times 10^{17}$ molec cm$^{-2}$ (7.9 %) in HDO, and −20‰ (9.7 %) in a posteriori $\delta D$. Although the absolute bias in $\delta D$ is higher for cloudy scenes than for clear-sky scenes, the relative bias is not. This is connected to different conditions in cloudy and clear-sky weather. The distributions of the differences (TROPOMI − TCCON, visualised by the violin plots in Fig. 8) vary

considerably between stations. Outliers are present, which shows that statistics over an adequate amount of data is needed for interpretation. Altogether, the performance of the new scattering retrieval for clear-sky scenes is similar to the one of the non-scattering retrieval, even though the scattering retrieval yields much more data. Biases are slightly smaller in HDO but slightly larger in a posteriori $\delta D$.





**Figure 8.** Bias in $H_2O$ columns (**a**), relative bias in $H_2O$ columns (**b**), bias in HDO columns (**c**), relative bias in HDO columns (**d**), bias in $\delta D$ (**e**), relative bias in $\delta D$ (**f**) for clear-sky scenes (blue), cloudy scenes (red) and the non-scattering retrieval (green). The violin plots visualise the distributions of differences between TROPOMI and TCCON, the boxplots mark quartiles and the dashed lines inside the boxes the mean. Coloured horizontal lines denote station-to-station medians and the shading around them the station-to-station quartiles.





For a direct comparison of the new scattering retrieval to the non-scattering retrieval by Schneider et al. (2020), only
ground pixels for which both retrievals yield valid data are considered. The distributions of the differences (TROPOMI −
TCCON) is very similar at most stations, with significant differences only at the coastal stations Burgos and Wollongong
and at Park Falls. The station-to-station median bias for this scene selection at low and middle latitude stations is in $H_2O$
$4.4 \times 10^{20}$ molec cm$^{-2}$ or 0.3 % for the scattering retrieval vs. $-4.2 \times 10^{18}$ molec cm$^{-2}$ or 0.4 % for the non-scattering retrieval
and in HDO $-4.5 \times 10^{16}$ molec cm$^{-2}$ or $-1.1$ % vs. $-9.3 \times 10^{16}$ molec cm$^{-2}$ or $-1.3$ %. In a posteriori $\delta D$ it is $-14$‰ (7.5 %)
for the scattering retrieval vs. $-11$‰ (5.4 %) for the non-scattering retrieval. This demonstrates that the performance of both
retrievals is comparable.

## 4.2  High-altitude stations

Ground stations on high mountains are special because the station height and the mean surface altitude of co-located satellite
ground pixels typically differ considerably, which means that different air columns are observed by both. This leads to high
biases if not accounted for. Therefore, the chosen prior plays an important role in this situation. To demonstrate the role of the
prior in potential corrections, an additional run with HDO prior profiles obtained by an assumed more realistic $\delta D$ profile as
described in Sec. 2 has been performed. During the co-location, the same ground pixels are considered for both runs. Moreover,
averaging kernels are not applied for this analysis because the prior profiles of the retrieval are used for the altitude correction.

The left column of Fig. 9 demonstrates the high biases of uncorrected clear-sky observations near Zugspitze (2964 m a. s. l.),
which for the standard prior amount to 185 % in $H_2O$, 232 % in HDO and 75‰ in $\delta D$. Nevertheless, the time series does
follow the relative variability of the reference.

The ground station on top of the mountain is always higher than the (mean) ground pixel altitude. To correct for the altitude
differences, the partial columns of the TROPOMI observations above the station height are considered by truncating the scaled
profile of the retrieval at the altitude of the station. This is the same procedure applied by Schneider et al. (2018, Sec. 4). The
second column of Fig. 9 depicts the resulting time series. The bias in both $H_2O$ and HDO is greatly reduced to $-54$ % and
$-48$ % for the standard prior and $-55$ % and $-54$ % for the depleted prior. In a posteriori $\delta D$ a large difference between both
priors is visible: while the bias for the scaled prior is practically the same as for the uncorrected case, 73‰, it is largely reduced
to 4‰ for the depleted prior. The first is due to the fact that the altitude correction in $H_2O$ and HDO cancels out when dividing
HDO by $H_2O$ if the same profile shapes are used. On the other hand, the small bias in $\delta D$ in the second case shows that the
assumed depleted HDO profile shape is indeed a good estimate for this case.

Another possibility is to utilise the shielding of clouds. To this end, scenes with optically thick clouds at an altitude similar
to the station height as specified in Tab. 4 are selected. In these cases, the satellite measurement is sensitive above the cloud
but insensitive below the cloud. Figure 10 illustrates the corresponding averaging kernels for a clear-sky and a cloudy scene.
Since the FTIR has to see the sun and thus can measure only through gaps in the clouds or when the cloud cover changes
within the co-location time, the amount of data for cloudy scenes is very small. Thus, the co-location radius is extended to
$r_{90°} = 50$ km in this case. The inferred columns are corrected for the altitude difference between ground pixel and station height
as described above. The right panel of Fig. 9 depicts the resulting time series. The biases in the columns and in a posteriori $\delta D$

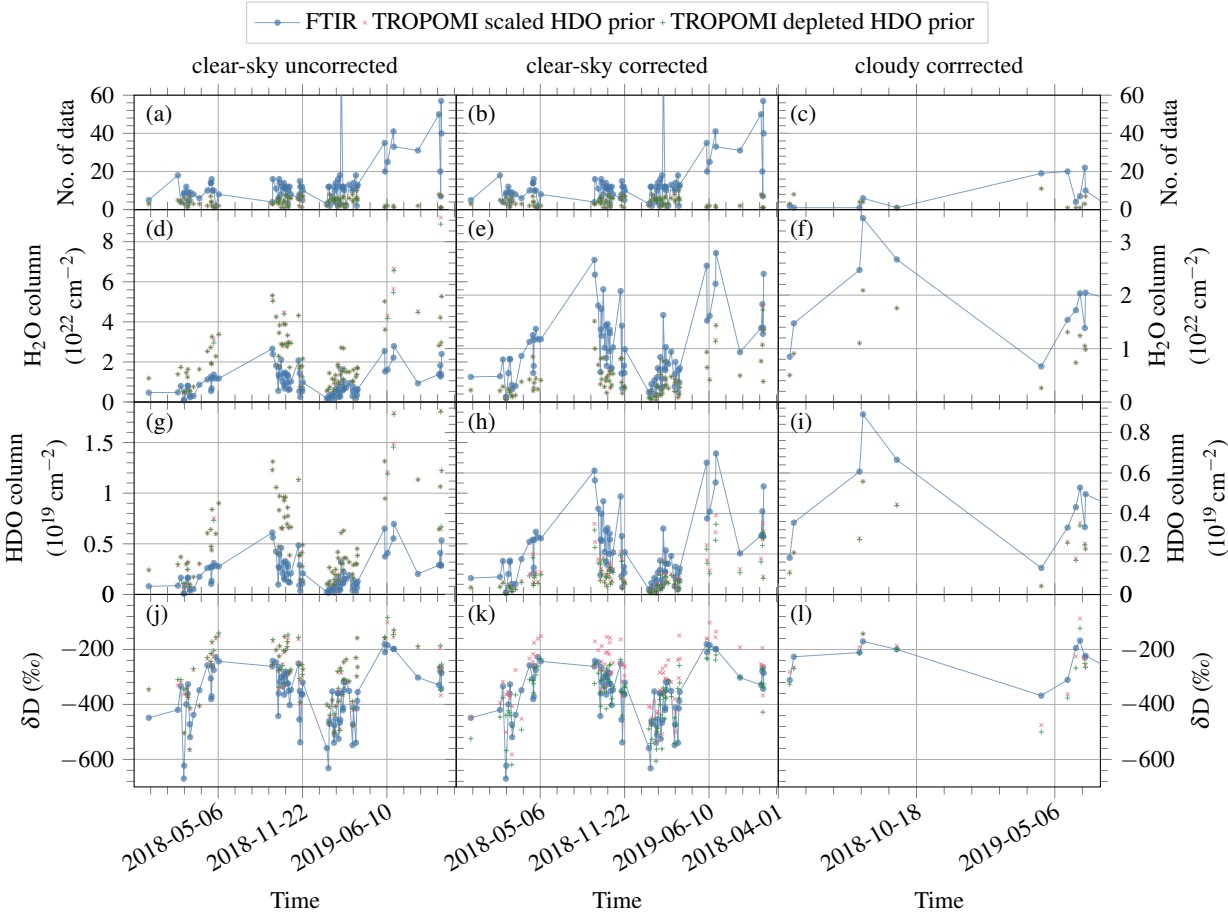

**Figure 9.** Time series of the amount of individual measurements per day near (first row), bias in $H_2O$ column (second row), bias in HDO column (third row), and bias in $\delta D$ (fourth row) at the high-altitude station Zugspitze (2964 m a. s. l.). The left panels **(a)**, **(d)**, **(g)** and **(j)** show clear-sky measurements without altitude correction; the centre panels **(b)**, **(e)**, **(h)** and **(k)** show the same measurements with altitude correction; and the right panels **(c)**, **(f)**, **(i)** and **(l)** show observations over optically thick clouds within an altitude range 1000 m above and 500 m below the station height. Please note that in the left panels the $H_2O$ and HDO axes are different than in the centre and right panels, as indicated by the axis ticks.

**Table 4.** Filter criteria for cloudy-sky scenes at high-altitude stations. Here $h_s$ denotes the height of the ground site.

| Quantity | Filter |
|---|---|
| | **Filter for cloudy scenes** |
| Cloud height | $h_s - 500\,\mathrm{m} \leq h_{cld} \leq h_s + 1000\,\mathrm{m}$ |
| Cloud optical thickness | $\tau_{cld} \geq 2$ |





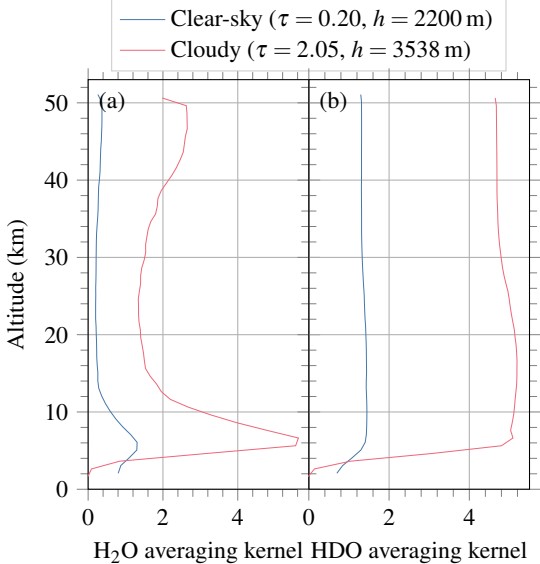

**Figure 10.** Averaging kernels of $H_2O$ **(a)** and HDO **(b)** for a clear-sky scene (orbit 4725 on 11 Sep 2018, blue) and a cloudy scene (orbit 4839 on 19 Sep 2018, red) near Zugspitze.

are acceptable for both priors. They amount to 4‰ for the scaled prior and $-24$‰ for the depleted prior. That the shielding yields good agreement with the scaled prior shows that the data provides information about the vertical distribution.

Figure 11 depicts biases for both high-altitude stations Zugspitze and Izaña. It confirms the behaviour seen in the time series at Zugspitze for both stations. Uncorrected clear-sky observations yield a large bias in all quantities. The altitude correction greatly reduces the bias in the $H_2O$ and HDO columns. In $\delta D$, the correction cancels out when assuming the same vertical distributions of $H_2O$ and HDO so that the bias remains. However, the altitude correction with a realistic prior yields a substantial reduction of the bias in $\delta D$. For cloudy scenes with optically thick clouds in similar altitudes than the station height, the biases

are also relatively small, although the validation is hampered by a small amount of data.

### 4.3 MUSICA-NDACC

Recent MUSICA-NDACC data are available for two low-altitude stations. Karlsruhe is also in the TCCON network so that a comparison is possible. MUSICA-NDACC provides fewer measurements than TCCON (113 vs. 170 for clear-sky scenes and 83 vs. 148 for cloudy scenes). This is, among others, due to longer duration of individual FTIR measurements for NDACC

compared to TCCON. Correlations, as shown in Figure 12, are excellent in the retrieved columns. For clear-sky scenes, Pearson coefficients are 0.98 in $H_2O$ and 0.99 in HDO, the same numbers as for TCCON (compare Fig. 6). For cloudy scenes, correlations with MUSICA-NDACC are with 0.98 in $H_2O$ and 0.99 in HDO even better than with TCCON, however with considerably less data points. A posteriori $\delta D$ also has excellent correlation coefficients of 0.93 for clear-sky scenes and 0.91 for cloudy scenes, which is better than with TCCON. The bias for clear-sky scenes is $1.8 \cdot 10^{21}$ molec cm$^{-2}$ (2 %) in $H_2O$,



**Figure 11.** Biases for high-altitude TCCON stations plotted similarly as in Fig. 8, but for retrievals with the standard scaled HDO prior profile (blue) and a HDO prior profile obtained by assuming a more realistic $\delta D$ profile described in Sec. 2. Shown are **(a)** the number of days with observations, **(b)** the bias in $H_2O$ columns, **(c)** the relative bias in $H_2O$ columns, **(d)** the bias in HDO columns, **(e)** the relative bias in HDO columns, **(f)** the bias in $\delta D$, and **(g)** the relative bias in $\delta D$. For each station, three entries are shown which correspond to uncorrected clear-sky observations, clear-sky observations corrected for the station altitude and altitude-corrected cloudy observations.



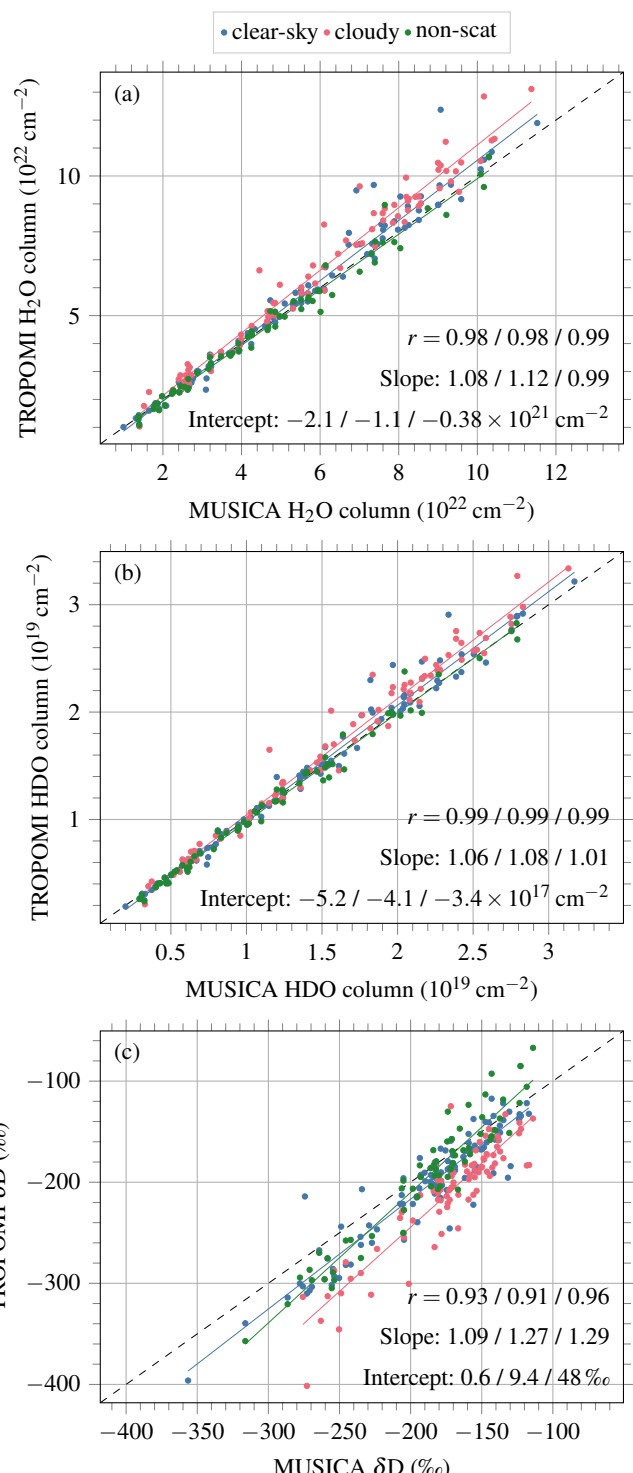

**Figure 12.** Correlations of TROPOMI observations against corrected MUSICA-NDACC measurements of $H_2O$ columns **(a)**, HDO columns **(b)** and a posteriori $\delta D$ **(c)** at Karlsruhe.



$2.5 \times 10^{17}$ molec cm$^{-2}$ ($-0.1\%$) in HDO, and $-16\%$ ($8.4\%$) in $\delta$D. For cloudy scenes, the bias is $6.4 \times 10^{21}$ molec cm$^{-2}$ ($9.9\%$) in $H_2O$, $9.3 \times 10^{17}$ molec cm$^{-2}$ ($4.8\%$) in HDO, and $-37\%$ ($21\%$) in $\delta$D. This is significantly larger than for TCCON.

Only one other low-altitude station provides MUSICA-NDACC data with temporal overlap with the TROPOMI mission, namely Kiruna. This is a high-latitude station, so that high biases are expected. They amount to $2.6 \cdot 10^{21}$ molec cm$^{-2}$ ($4.6\%$) in $H_2O$, $1.6 \cdot 10^{17}$ molec cm$^{-2}$ ($-3.5\%$) in HDO, and $-58\%$ ($24\%$) in $\delta$D for clear-sky scenes and $5.0 \cdot 10^{21}$ molec cm$^{-2}$

($12\%$) in $H_2O$, $6.4 \cdot 10^{17}$ molec cm$^{-2}$ ($5.1\%$) in HDO, and $-51\%$ ($23\%$) in $\delta$D for cloudy scenes. With only two stations, it is not meaningful to make statistical statements.

### 4.4 WISPER aircraft measurements over the ocean

In order to validate the retrievals over oceans, aircraft profiles from the ORACLES field campaign in 2018 are used as reference. The data reduction method is described in Sec. 3.3.

Figure 13 shows a time series of total columns computed from aircraft profiles and co-located TROPOMI retrievals over the North Atlantic ocean. The bias is $(-6.1 \pm 11) \times 10^{21}$ molec cm$^{-2}$ or $(-3.9 \pm 6.9)\%$ in $H_2O$ and $(-3 \pm 15)\%$ in $\delta$D. The validation over the ocean is hampered by very few data points. Nevertheless, the comparison to the aircraft profiles shows that the performance of the retrieval over the ocean is good.

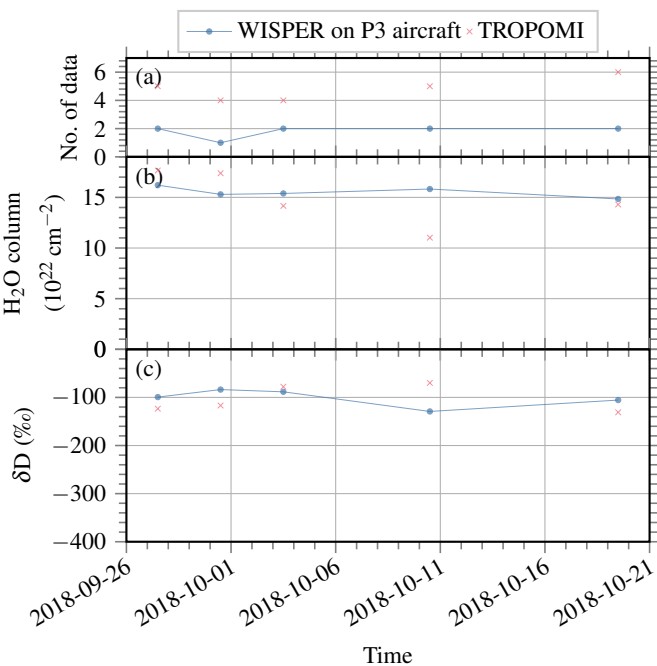

**Figure 13.** Times series of **(a)** number of measurements, **(b)** daily averaged total columns of $H_2O$ and **(c)** $\delta$D from aircraft profiles (blue) and co-located TROPOMI retrievals (red).


# 5 Demonstration of applications of the data set

## 5.1 Global picture

Figure 14 demonstrates a global picture of the new data set with a monthly average for September 2018. The most prominent improvement compared to the plot of the non-scattering product shown in Schneider et al. (2020, Fig. 10) is a huge enhancement

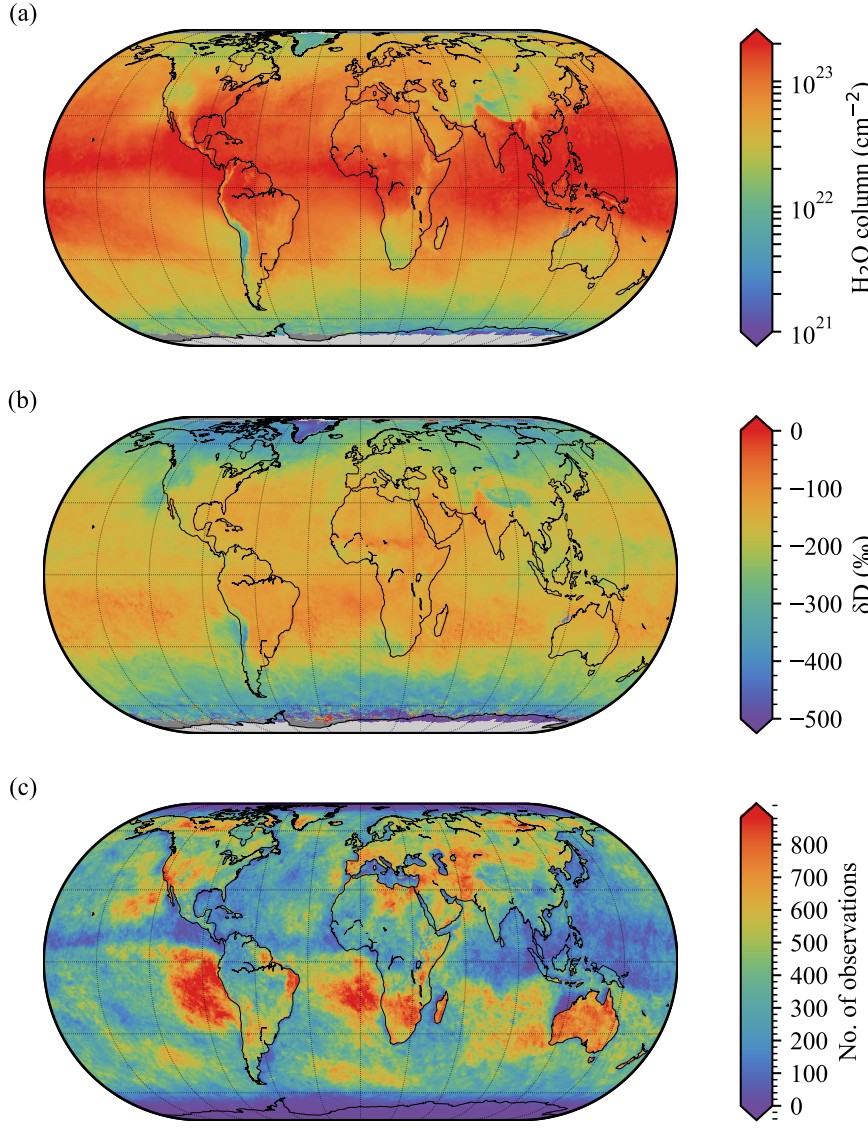

**Figure 14.** Global plots of **(a)** average $H_2O$, **(b)** average $\delta D$, and **(c)** number of observations for September 2018 on a $0.5° \times 0.5°$ grid. The average of $\delta D$ is weighted with the $H_2O$ column.





in data coverage, most prominently over the oceans and in regions at low latitudes with persistent clouds (e. g. over the Amazon, Central Africa and Oceania), where the non-scattering retrieval yields no data. Near these regions and also over northern India,

$\delta D$ is lower than in the clear-sky only data product which is attributed to different weather conditions at cloudy days compared to clear-sky days.

In the spatial distribution of column integrated $H_2O$ and $\delta D$ shown in Fig. 14 the major characteristic features of the atmospheric water cycle and the isotopic effects as described by Dansgaard (1964) can be recognised. In Fig. 14a, a very moist intertropical band can be distinguished. The typically deep convective clouds associated with the moistest band shield

most of the scenes in the area of the intertropical convergence zone (Equatorial Atlantic, and Pacific, see Fig. 14c). This might lead to a slight underestimation of the total column $\delta D$ in this area. The moist continental regions associated with the northern hemisphere's fading summer monsoon systems (West African Monsoon, southeast Asian Monsoon) show relatively more depleted $\delta D$ total columns (see coastal West Africa, China) compared to other regions in the same latitudinal band (see subtropical oceans, India and North Africa). The combined ($H_2O$, $\delta D$) information is likely to provide more insight into mixing

and cloud and below cloud evaporation effects in these regions (Noone, 2012).

The subtropical ocean regions are all associated with relatively high total column $\delta D$ except for spots with distinctly lower values found along the eastern coasts of the ocean basins. In the latter coastal regions around $30°$ N/S stratocumulus capped areas with strong inversions are prominent and enhanced subsidence is frequently observed (Norris, 1998; Myers and Norris, 2013), which probably leads to the distinct local minima in total column $\delta D$ and total column $H_2O$.

In the regions of frequent occurrence of extratropical cyclones (storm tracks) over the midlatitude western North Atlantic, western North Pacific and in the Southern Ocean, sharp gradients can be observed in total column $\delta D$. The equatorward flanks of the storm tracks are associated with warm air and total column $\delta D$ of about $-100‰$. In contrast, much lower values of about $-300‰$ can be observed on the subpolar flanks of the storm tracks. In these regions the frequent occurrence of extratropical cyclones (Wernli and Schwierz, 2006) strongly modulate the variability of $\delta D$ signals of atmospheric water vapour (Thurnherr

et al., 2020b). In the Southern Ocean, the spatial pattern of oceanic total column $\delta D$ reflects the spiral shaped winding of the Southern Ocean storm track around Antarctica with more frequent storms in the central South Atlantic compared to the Central South Pacific. In the latter region a tongue of more enriched total column $\delta D$ reaches far south towards the Antarctic coast.

Along the Antarctic coast, the very low total column $\delta D$ might be due to cold air outbreaks or strong katabatic outflows at low levels, advecting very depleted Antarctic water vapour over the ocean (Thurnherr et al., 2020b). However, the data

availability over these coastal Antarctic region is very limited (Fig. 14c). A comparison with high resolution isotope-enabled numerical model simulations in these regions as well as in very high altitude mountainous regions, would certainly help in ruling out important biases due to uncertainties associated with the retrieval in these regions with complex topography and small-scale variations in the albedo.

The data coverage, as can bee seen on the example for the month of September 2018 in Fig. 14c, is highly variable in space.

Particularly over tropical oceanic regions, the data is very sparse due to shielding by high clouds. Over high latitude land regions, the data is also sparse due to high solar zenith angles and low surface albedos (recall the SZA filter and albedo filter,





cf. Tab. 1). In contrast, particularly in regions of enhanced subsidence in the subtropics a large number of observations are available. A weak seasonal cycle in the amount of observations exists particularly at high latitudes.

## 5.2 Single overpasses

Figure 15 demonstrates single overpass results over the North Atlantic Ocean. On 17 January 2020 a cold air outbreak forms along the North American east coast, behind a cold front associated with a North Atlantic cyclone. The cold front can be identified in Fig. 15a by the quasi-zonal cloudy band, marked by a strong gradient of low to high total column $H_2O$ between $15°$ N and $25°$ N across the front. The cold air mass (see low values of potential temperature at 850 hPa behind the cold front in Fig. 15f) travels southward towards the tropics between 17 and 20 January 2020 (Figs. 15–17). The cold, subsiding air behind

the cold front is very dry (Fig. 15a) and is associated with low total column $\delta D$ values between $-400$ and $-200‰$ (Fig. 15b) which are characteristic of the cold sector of extratropical cyclones (Thurnherr et al., 2020a). Marine cold air outbreak clouds are typically low level clouds with high cloud fraction (stratocumulus, cumulus, Fig. 15e) and moderate optical thickness (Fig. 15c, Fletcher et al., 2016). The very high $\delta D$ values of $\sim 0‰$ stretching in a bow from $\sim 20°$ N, $40°$ W westward are caused by low sensitivity in low altitudes due to cloud shielding. These sensitivity issues are reflected by very low values of the

column averaging kernel (Fig. 15d). The magnitude of the null-space error is determined by the deviation of the shape of the prior profile to the real profile. The prior depends on time and location, thus the null-space error may be different in different regions. Nevertheless, these data still contain valuable information that can be interpreted in combination with measurements or model simulations providing vertical profiles of $H_2O$ and HDO that can be combined with the vertical sensitivity of the satellite retrievals.

The analysis of successive overpasses between 18 and 20 January (Fig. 16, 15, 17) shows a rapid moistening of the originally very dry and depleted cold air mass. When it leaves the North American continent on 18 January the cold sector air has total column $\delta D$ of less than $-400‰$. On 20 January, when the cold front reaches into the tropics, the $\delta D$ of the cold sector is in the range $-300$ to $-200‰$. The dry and cold air subsiding above the boundary layer typically induces large humidity gradients near the ocean surface and consequently leads to enhanced surface evaporation fluxes that favour a rapid moistening

and continuous increase in $\delta D$ of cold sector air as it travels southward (Aemisegger and Papritz, 2018). The $\delta D$ in Fig. 15b shows large spatial variability in the cold sector hinting towards different degrees of vertical mixing in different regions of the cold sector, most likely due to variations in subsidence strength. The latter aspect could be investigated in more detail using this dataset in combination with a numerical weather model including isotopes.

This variability in $\delta D$ at low total column $H_2O$ can also be observed when displaying the cold sector data in a ($H_2O$, $\delta D$)

phase space (Fig. 18). In contrast to the cold air mass behind the cold front, the trade wind air mass in front of the cold front is associated with very high total column $\delta D$ (Fig. 18b). Reduced subsidence and stronger shallow convective activity with deeper clouds are the reason for the higher $\delta D$ on the warm, trade wind side of the front (see also Aemisegger et al., 2020, for a discussion on the impact of extratropical intrusions behind cold fronts on the low-level $\delta D$ signals in the tropics).

In future comparisons of TROPOMI all-sky observations with vertical profiles from aircraft-based measurement campaigns

will be helpful for identifying potentially remaining biases in very dry compared to very moist conditions. Furthermore, studies







**Figure 15.** TROPOMI single overpass results of $XH_2O$ **(a)**, $\delta D$ **(b)**, retrieved effective cloud optical thickness **(c)** and column averaging kernel at the surface **(d)** over the North Atlantic on 19 Jan 2020; ERA5 cloud fraction **(e)** and ERA5 potential temperatures at 850 hPa at 15:00 UTC **(d)**. The grey contours in all panels show ERA5 mean sea-level pressure at 15:00 UTC with a contour line distance of 2 hPa. The black contours in (f) show vertical winds at 500 hPa in levels of $0.5\,\mathrm{Pa\,s^{-1}}$. The boxes in (a) and (b) mark the regions for which Rayleigh plots are depicted in Fig. 18.





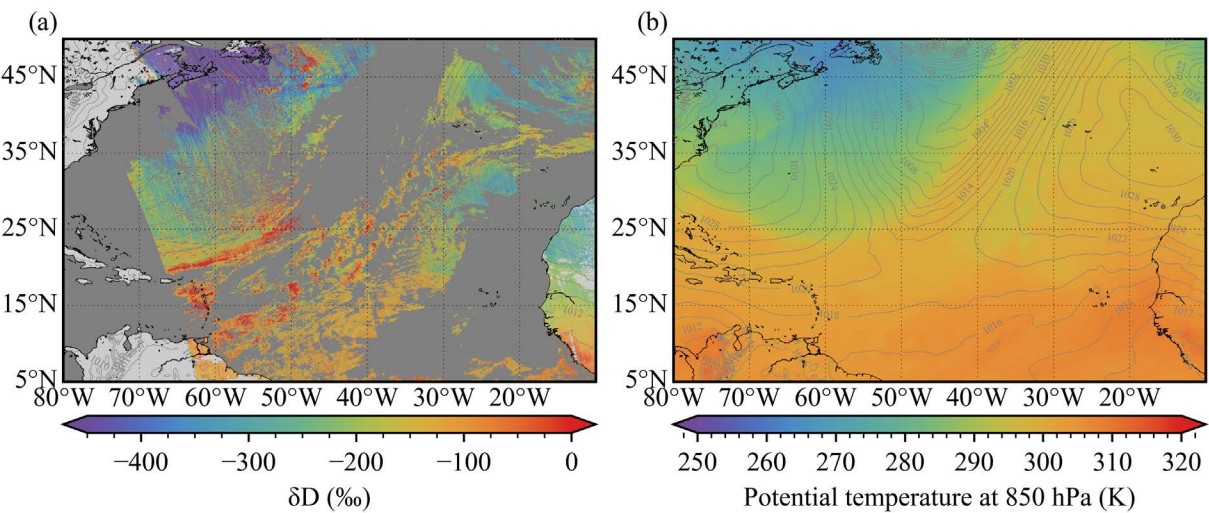

**Figure 16.** TROPOMI single overpass $\delta$D **(a)** and ERA5 potential temperatures at 850 hPa at 15:00 UTC **(b)** on 18 Jan 2020. The grey contours in all panels show ERA5 mean sea-level pressure at 15:00 UTC with a contour line distance of 2 hPa.

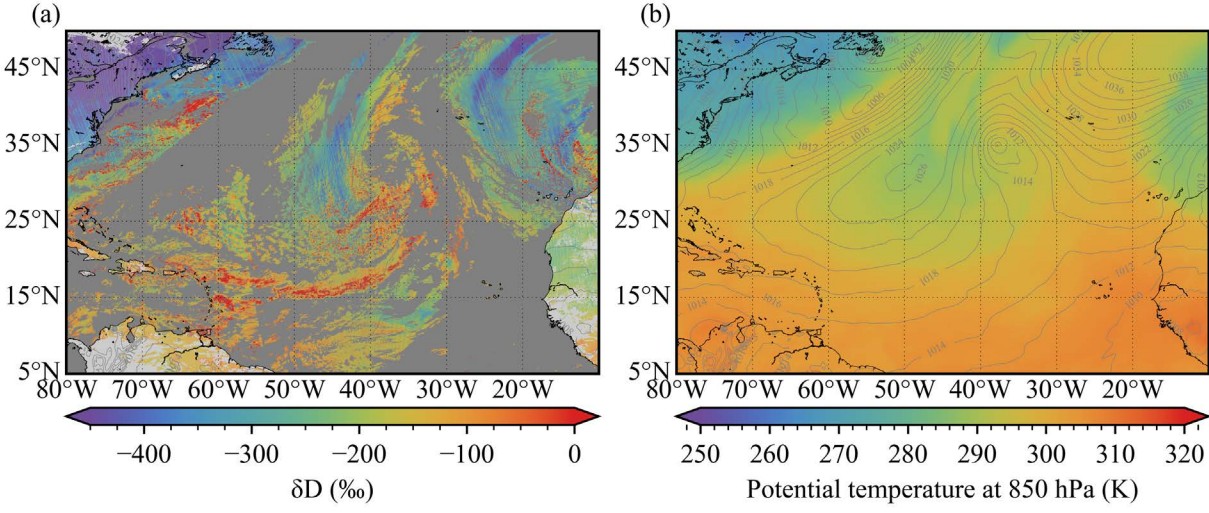

**Figure 17.** TROPOMI single overpass $\delta$D **(a)** and ERA5 potential temperatures at 850 hPa at 15:00 UTC **(b)** on 20 Jan 2020. The grey contours in all panels show ERA5 mean sea-level pressure at 15:00 UTC with a contour line distance of 2 hPa.





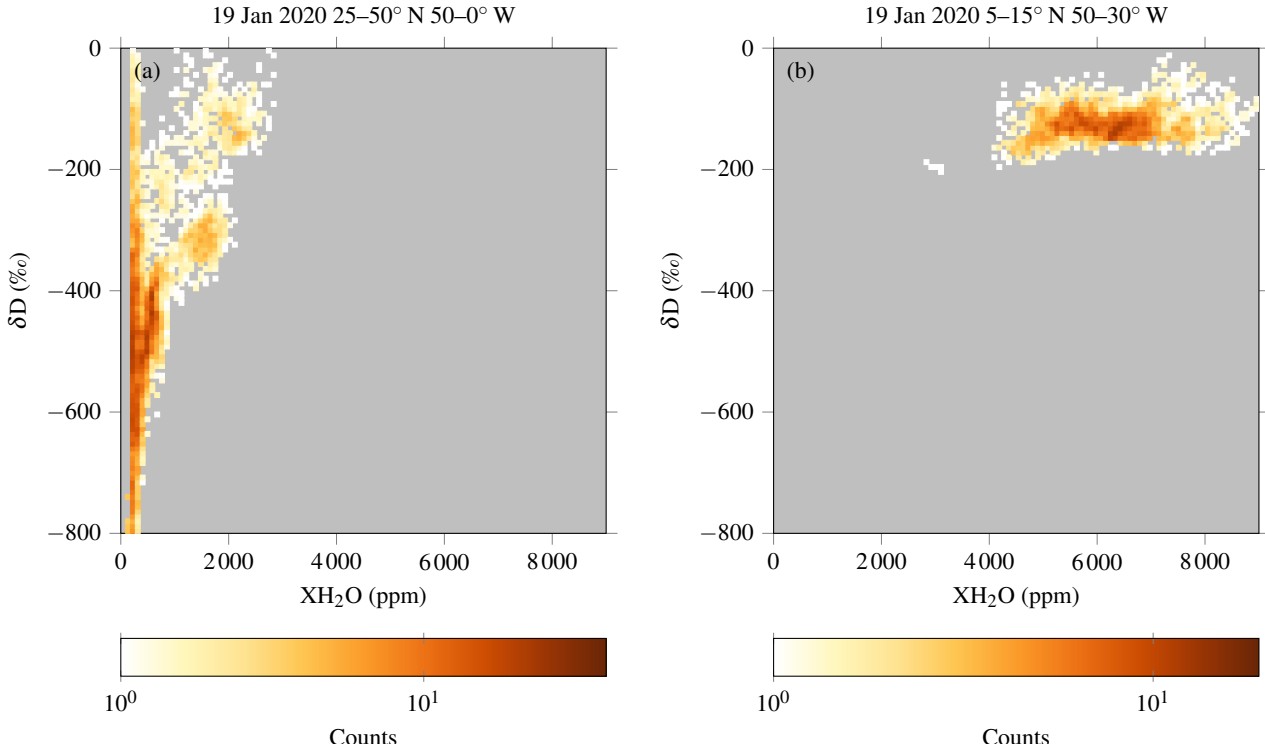

**Figure 18.** Histograms of TROPOMI observations on 19 Jan 2020 **(a)** in the area 25–50° N, 50–0° W comprising the cold sector and **(b)** in the area 5–15° N, 50–30° W containing the cold front.

combining TROPOMI data with high resolution numerical modelling will provide a promising data basis for studying the interaction between the moist boundary layer and the subsiding dry free tropospheric air, which is key in determining the variability in the low-level cloud cover properties.

## 6 Summary and conclusions

This work presents a new data set of $H_2O$ and HDO columns over cloudy or clear-sky scenes retrieved from TROPOMI short-wave infrared measurements. Effective cloud parameters are fitted in the spectral window 2310 nm to 2338 nm and taken over to the final fit of the trace gases in the spectral window 2354.0 nm to 2380.5 nm. Surface albedos are regularised to the one-year average of the non-scattering retrieval by Schneider et al. (2020).

The performance of the new retrieval is similar to that of the non-scattering retrieval when comparing the same ground
pixels, i. e. clear-sky scenes over land. Nevertheless, the scattering retrieval yields much more data, even for scenes classified as clear-sky since the filtering is less strict. The median bias to TCCON at low-altitude stations in low and middle latitudes is for clear-sky scenes $1.3 \times 10^{21}$ molec cm$^{-2}$ (1.8 %) in $H_2O$ columns, $2.0 \times 10^{16}$ molec cm$^{-2}$ (−0.3 %) in HDO columns and





$-8‰$ (4.6 %) in a posteriori $\delta$D, the one for cloudy scenes is $4.7 \times 10^{21}$ molec cm$^{-2}$ (8.8 %) in $H_2O$, $1.0 \times 10^{18}$ molec cm$^{-2}$ (6.5 %) in HDO columns and $-20‰$ (12 %) in $\delta$D. At high latitudes, the bias is higher (up to about 20 % in the columns and 40‰ in a posteriori $\delta$D) due to difficult measurement geometries with typically high solar zenith angles and low surface albedos meaning low signal-to-noise ratios.

At high-altitude stations, the altitude difference between satellite ground pixel and FTIR instrument has to be taken into account. If not corrected for, different partial columns are compared which leads to high biases. A correction by taking the partial column of the satellite observation above the ground station height largely reduces the biases in the $H_2O$ and HDO columns, however the bias in a posteriori $\delta$D remains because the correction cancels out when using the same profile shapes. This bias can be eliminated by using the shielding of clouds: for cloudy scenes with cloud height similar to the station height, the bias in a posteriori $\delta$D is very low. This shows that the shielding by clouds provides information about the vertical distribution. For clear-sky observations, the bias in $\delta$D can be eliminated by using more realistic profile shapes for HDO: an experiment with a prior profile of HDO computed from an assumed more realistic profile of $\delta$D shows a low bias in a posteriori $\delta$D after the altitude correction.

Over oceans, the retrievals are validated with aircraft profile measurements from 2018. Although the validation is hampered by a limited amount of reference measurements, the good retrieval performance is confirmed.

The amount of data in the new data set is tremendously increased compared to the non-scattering retrieval by Schneider et al. (2020). Besides more data for clear-sky scenes over land due to less strict filtering, retrievals over low clouds give new insights, particularly over oceans where the non-scattering retrieval cannot yield data. Single overpasses yield meaningful results which enables new case studies. As an example with cloudy scenes over the oceans, a cold air outbreak in January 2020 is shown. Retrievals from consecutive days nicely show the transport of depleted continental air from high to subtropical latitudes.

More reference measurements over oceans, either aircraft or ship based, will be useful to complement the validation. Furthermore, a calibration of the TCCON HDO product would be beneficial. Moreover, a homogenisation of the ground-based data products by TCCON and MUSICA-NDACC would be valuable.

*Data availability.* The TROPOMI HDO data set of this study is available for download at ftp://ftp.sron.nl/open-access-data-2/TROPOMI/
tropomi/hdo/10_3/. TCCON data are available from the TCCON Data Archive at https://tccondata.org/. MUSICA data are available from
ftp://ftp.cpc.ncep.noaa.gov/ndacc/MUSICA/ or from https://doi.org/10.5281/zenodo.48902 (Barthlott et al., 2016). Aircraft-based WISPER
data from the ORACLES 2018 campaign are available from https://espoarchive.nasa.gov/archive/browse/oracles/P3/mrg1.

*Author contributions.* AS prepared the manuscript with contributions from all co-authors. FA performed the case study in Sec. 5.2. DN and
DH provided aircraft data. RK provided TCCON data.



*Competing interests.* The authors declare that they have no conflict of interest.

*Disclaimer.* Plots/data contain modified Copernicus Sentinel data, processed by SRON.

*Acknowledgements.* This work was supported by the ESA Living Planet Fellowship project Water vapour Isotopologues from TROPOMI
(WIFT). The TROPOMI data processing was carried out on the Dutch national e-infrastructure with the support of the SURF Cooperative.
Kimberly Strong, Justus Notholt, Debra Wunch, Christof Petri, Nicholas Deutscher, Frank Hase, Yao Té, Thorsten Warneke, Ralf Sussmann,
Paul Wennberg, Isamu Morino, Laura T. Iraci, Kei Shiomi, Matthias Schneider, David Griffith, and Dave Pollard provided TCCON data.
Matthias Schneider provided MUSICA-NDACC data.



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
