# Peer review of "Retrieving H2O/HDO columns over cloudy and clear-sky scenes from the Tropospheric Monitoring Instrument (TROPOMI)"

_Atmospheric Measurement Techniques, 2021_

## Author Response (AR1)

**Review by Anonymous Referee #1**

*The paper by Schneider et al. presents an extension of the former TROPOMI column data set published by the authors last year that was restricted to cloud-free scenes. It is great that the authors now can also retrieve the cloudy scenes and thus extend their previous data set. This allows now for a much wider scope of scientific application than the previous data set and presents therefore a quite valuable data set.*

Thank you for your review and your positive evaluation of our work. In the following, all individual comments are quoted in italics and our response is given below.

*I think the only drawback is that the structure of the paper is quite similar to the previous paper by the authors (Schneider et al., 2020). It would have been nicer and maybe more interesting to set the focus on the comparison of the old and new data set and the additional gain and application possibilities of the new data set instead of just extending the previous analyses with some more stations (though this of course is also nice and valuable).*

The new data set (scattering retrieval) is independent of the old one introduced in the previous paper (non-scattering retrieval) and thus requires a separate validation. For comparison, we have additionally included the performance of the old data set. The direct comparison for the same scenes found at the end of Section 4.1 in the discussion paper has now been extended into a separate subsection and supplemented with a plot which we had left out in the initial submission due to the length of the paper.

*I have following comments I would like to ask the authors to consider before publication.*

- *P1, L1: The first sentence of the abstract is not clear without knowing what you have done. I would suggest rephrasing and being more precise.*

  We have rephrased this sentence as follows: "This paper presents an extended scientific HDO/$H_2O$ column data product from short-wave infrared (SWIR) measurements by the Tropospheric Monitoring Instrument (TROPOMI) including clear-sky and cloudy scenes."

- *P1, L4: Clouds are usually present over the oceans and over land. How does it then comes that you in case of cloud-free scenes derive data over land and if you consider cloudy scenes over oceans?*

  Cloud-free scenes can be retrieved over land only because bodies of water are too dark in the short-wave infrared to retrieve. Cloudy scenes are retrieved over both oceans and land. This is explained in the main text, but we have added a half-sentence in the abstract to also mention it there. The sentence now reads: "... particularly enabling data over oceans as the albedo of water in the SWIR spectral range is too low to retrieve under cloud-free conditions."

- *P2, L36–38: Same as for P1, L4. An explanation or more information would be quite helpful.*

  As explained in L31–32, the albedo of water in the short-wave infrared is too low to retrieve in cloud-free conditions over oceans. To make more clear that cloudy scenes are used over both oceans and land, the sentence in L38–40 is rephrased as follows:

  "This can be remedied by also considering scenes over low clouds, which enables data over oceans and greatly extends coverage over land. To this end, an updated retrieval is employed which accounts for scattering and estimates effective cloud parameters additionally to the trace gases."

- *P2, L40–41: This sentence is not easy to understand, please rephrase.*

  We have rephrased this sentence as follows: "Any loss of sensitivity to the partial column below the cloud is reflected by the column averaging kernel."

- *P2, L42: "Section 2....." this comes a bit suddenly. Add an introductory sentence to begin this paragraph of paper structure description as e.g. "The paper is structured as follows:" or write "In the next section we describe the retrieval set-up...."*

  We have rephrased it to "The next section describes the retrieval setup ..."

- *P3, L59 and 61: fitted to what? Please be more precise.*

We have rephrased this as follows: "The inversion derives the target trace gases $H_2O$ and HDO together with the interfering species $CH_4$ and CO and a Lambertian surface albedo from the observed spectrum in the spectral window from 2354.0 nm to 2380.5 nm (Scheepmaker et al., 2016). The isotopologue $H_2^{18}O$ is included in the forward model but not estimated in the inversion (i. e. the abundance is fixed at the prior value)."

- *P4, L99: "except if the data are assimilated using averaging kernels" I would suggest to make an extra sentence since also the data of all scenes can be used when the averaging kernels are applied. This does not only hold for data assimilation.*

We start a new sentence after "recommended to be taken into account by the user" as follows: "If averaging kernels are taken into account, e. g. when assimilating the data, all scenes can be used, although shielding by high clouds may result in quite low information content."

- *P6, L101: Retrieve what exactly? A specific gas or trace gases in general? Also here I would suggest to be more precise.*

We mean trace gases in general. We have rephrased the sentence as follows: "Such a surface albedo filter is not applied to cloudy scenes because clouds usually have high reflectivity, which allows the retrieval algorithm to work over very low surface albedos with high signal-to-noise ratio."

- *P6, L103ff: How is it guaranteed that the missing data that is filled in is reliable? Is there another filter used to check the quality or to filter out the non realistic filled up information?*

The a priori profile, which is obtained from the ECMWF analysis product, is the best estimation of the truth that is available. Under cloudy conditions there will always be a null-space error. The restriction to low clouds (cloud height filter) limits that error. To completely prevent such errors, the user needs to take the averaging kernel into account to avoid null-space errors.

- *P6, 110: How was this correction factor derived? By comparing FTIR to TROPOMI? If yes, how can you then be sure that the correction factor holds for both data sets (old and new)?*

The correction factor is determined by comparing TCCON to MUSICA-NDACC for instruments in both networks, since MUSICA $\delta$D is validated by aircraft measurements but TCCON HDO is not verified. The correction thus depends on FTIR measurements only and in particular is independent of the satellite retrieval. We have added a short summary on how the correction factor is determined, see below.

- *P6, L111: missing calibration of what? Be more precise.*

TCCON HDO misses an aircraft correction factor. That corrects systematic biases due to uncertainties in the spectroscopy which tend to be highly reproducible. Usually, the aircraft correction factor is determined by a comparison to aircraft measurements at TCCON sites, but no such reference measurements of HDO exist. We have changed the manuscript as follows:

"This factor accounts for a missing aircraft correction factor of TCCON HDO. The aircraft correction factor corrects systematic biases due to uncertainties in the spectroscopy which tend to be highly reproducible (Wunch et al., 2015). It is usually obtained from a comparison to airborne reference measurements at TCCON sites, however such measurements are lacking for HDO. Thus, Schneider et al. (2020) determined an effective factor by fitting TCCON a posteriori $\delta$D to MUSICA-NDACC $\delta$D because MUSICA-NDACC $\delta$D is validated with aircraft measurements."

- *P6, L127: Also in this paragraph you could be a bit more precise. So it seems the altitude difference is generally a problem and as higher the station as higher the problem/error gets? You could e.g. more clearly state here that this is the reason why a high-altitude stations are considered separately.*

To explain this better, the end of this paragraph now reads: "High-altitude stations are typically located on mountains and thus most co-located ground pixels have significantly lower surface height. Therefore, such stations are treated separately in Sec. 4.3."

- *P7, L135: Extended to 0 hPa? This is really high. Is this realistic? Are measurements made that high up?*

  0 hPa is the top of the layering of the forward model. To explain that, we have added the following half-sentence: "…to match the layering of the forward model." Since the abundance of water vapour is very low in these high altitudes, the contribution to the total column is very small, thus this choice influences the result very little.

- *Figure 4: Why using an average? Why is not the correction factor for each station used? How large is the introduced error by using an average?*

  The correction factor can only be determined at the few stations that are in both networks TCCON and NDACC. Most stations are only in one network, thus it is not possible to determine the correction factor individually at each station. We show that the difference between stations is small. Therefore, it is okay to use the average of the stations in both networks at all stations. To clarify this, we have appended the following sentence to the paragraph: "This correction is applied to all MUSICA-NDACC stations, i. e. also those not in the TCCON network."

- *P10, L160: Why should one fill the null-space with MUSICA-NDACC profiles if this data set is then used for validation? TROPOMI would then not be an independent data set. However, for validation rather an independent data set should be used.*

  The reason for such a potential filling of the null-space would be to account for limited sensitivity of the retrieval, however the prior profiles of MUSICA-NDACC are not realistic enough to do that.

- *P10, L173: This is a very complicated sentence and very hard to follow. Please rephrase and distribute the content over several sentences.*

  The sentence is rephrased as follows: "In order to derive total columns, the aircraft profiles are extended to the ground by assuming a constant mixing ratio equal to the lowest observed value and extended to the top with the scaled prior profile. These extended profiles are then vertically integrated to obtain total columns."

- *P11, L181: add "which is then used for the validation"*

  Added at the end of Section 3.3.

- *P12, Figure 5: The figure could be improved by using lines and symbols that are a bit thicker/larger. Looking at the number of data points it looks like there are almost no data points from TROPOMI.*

  The symbols are now larger, the colormap has been changed.

  A TCCON scan takes $2 \times 78$ s, thus a TCCON station can take spectra every $\sim 3$ minutes under good conditions. Although the measurement schedule varies by station, there can be a lot of measurements during the co-location time of 2 hours around a satellite overpass. In order not to confuse the reader, we now do not show the amount of TCCON measurements any more.

- *P13, Figure 6: Why is TCCON corrected? I thought MUSICA-NDACC is the data set that needs to be corrected?*

  Both TCCON and MUSICA-NDACC need corrections to resolves the inconsistency between both datasets in $H_2O$ and HDO. TCCON HDO is corrected so that TCCON $\delta D$ matches MUSICA-NDACC $\delta D$ because the latter is validated (see Section 3.1). MUSICA $H_2O$ (and HDO by the same factor to not change $\delta D$) is corrected because TCCON $H_2O$ is better validated than MUSICA-NDACC $H_2O$ (see Section 3.2).

- *P14, L228: Why is this connected to the different conditions in cloudy and clear-sky weather? Isn't this simply due to the nature of the relative differences? These are usually more severe (higher) for lower values. The differences of course (i.e lower values and larger differences) may be related to cloudy or clear-sky conditions.*

We do not completely understand this statement. We show in our validation that the relative bias in $\delta$D is comparable for clear-sky and cloudy scenes, but the absolute bias is higher for cloudy scenes than for clear-sky scenes. That means that the absolute $\delta$D values are systematically more negative in cloudy scenes, which we attribute to different weather conditions.

- *P16, L241: Refer here again to Figure 8 .Add also a table with the biases? The approximate values can be derived from the figure, but if one needs the exact value a table would be quite useful.*

This sentence does not refer to Figure 8, but to a separate comparison taking only the same ground pixels for both scattering and non-scattering retrieval into account. To make this difference more clear, we have made a separate subsection for this comparison.

The bias values at individual stations that can be taken from the plot are in our opinion precise enough, especially since there is a large spread in the differences (see violin and box plots), so that we deem no separate table necessary.

- *P17, Figure 9: As for Figure 5, the lines and symbols are too thin and the colors hard to differentiate. For the clear-sky data the corrected and uncorrected data are shown, but for cloudy only the corrected data are shown. Why? Shouldn't also here both been shown, corrected and uncorrected?*

The size of the symbols is increased and the colours changed.

In the cloudy case the retrieval is not sensitive to the partial column below the cloud and thus the station height and therefore is not reliable. The station altitude is significantly higher than the maximal cloud height for the cloudy scene filter.

- *P18, L278: What is the "scaled" and what the "depleted" prior?*

Not sure which place in the manuscript you are referring to. We have added the following sentences at line 247: "This prior is referred to as "depleted" prior because a depletion in HDO is assumed to compute it from the humidity profile. The standard prior is also referred to as "scaled" prior because it consists of a scaled humidity profile (i. e. corresponding to $0\,\permil$ $\delta$D)."

We have also added a sentence in the figure caption of Figure 9: "The red points correspond to the standard prior which is scaled from the humidity profile, while the green points correspond to the prior computed assuming a more realistic $\delta$D profile."

- *P18, Figure 10: Also here the figure could be improved by using a thicker line style.*

Line width is increased.

- *P21, L294:"The data reduction method is described in Sect. 3.3"? You mean the collocation criteria is described in Sec. 3.3?*

We mean not only the co-location criteria but also the computation of the total column from the aircraft profile with limited height coverage. We have now written "co-location method".

- *P21, Figure 13: Use also here colors that are better differentiable and line styles that are thicker and better visible.*

Symbol size is increased an colours changed.

- *P22ff: I really appreciate that you demonstrate the application of the data set, however, since your paper is already quite long and complex I wonder if it wouldn't be better to have an own, more sophisticated application paper where you also could include model simulations. You could only show here Figure 14 and give a short description of where exactly you gain more information and in which areas thus more sophisticated studies are possible. Another option would be to keep this example short. In that case, I would suggest skipping the details on Figure 14 (the discussion of the dD distribution, L307–L333, it does definitely not need to be that detailed) and put the focus only on the case study itself.*

The case study based on single overpasses over the ocean really demonstrates the novelty of this dataset, thus we consider it an important part of the paper. Therefore, we concentrate on that case study and skip the detailed description of the monthly mean global plot (L307–L333). To provide a concrete insight into the benefit of $\delta$D total column data compared to $H_2O$ alone in process-based studies of the atmospheric water cycle an additional figure with the distribution of the ($H_2O$, $\delta$D) pairs in the tropics is shown in Fig. 17. This figure is shortly discussed in Section 5.1.

- *P23, L305: Here you jump from data coverage to different dD amounts. Of course, different weather conditions result in different dD amounts, but that relation should be better explained.*

Yes, here we shortly attract the reader's attention to the fact that a comparison between the monthly $\delta$D distributions from the scattering and non-scattering retrievals is not trivial, because the monthly means result from sampling over different types of weather conditions, which makes a direct comparison over given regions difficult. We find this an important point and therefore kept this statement. The discussion between lines 307–333 in the first submission was exactly intended to explain how different weather situations can lead to differences in $\delta$D. However, we agree with the previous comment of the reviewer and therefore removed the latter text in the interest of keeping the paper focussed and short.

- *P23, L310: "most of the scenes in the area" is rather confusing and I would suggest to rephrase to "dominate the scenes".*

This paragraph has been removed.

- *P23, L311: "slight underestimation of total column dD" Why? Shouldn't the cloudy retrieval cover these values?*

This paragraph has been removed.

- *P27, L375: "or"? I though the cloudy data set contains all data? Or then do you use all data? This should be better explained throughout the paper.*

Here, "or" was not meant as exclusive or. The data set includes both clear-sky and cloudy scenes. We have changed "or" to "and" to avoid the misunderstanding.

- *General question: If a scientific study is done is then the cloudy data set enough or does one need both data sets (the cloudy and the non-cloudy data set)? Then it would be interesting to see how a combination of both data sets can be used.*

Do you refer to the scattering and non-scattering data sets? Only the scattering data set is needed for a scientific study and supersedes the old non-scattering data set. The new scattering data set contains both cloudy and clear-sky scenes, while the old non-scattering dataset contains only clear-sky scenes.

- *P28, L396–397: I am not sure if you are here a bit too optimistic. This could be just coincidence.*

We have rephrased this to "Although the validation is hampered by a limited amount of reference measurements, the available data shows a good retrieval performance."

The corresponding sentence in Section 4 is rephrased similarly.

**Technical corrections**

- *P1, L4: Add "new" so that it reads "The new data set......"*

Done.

- *P1, L12 and several other occasions in the manuscript: "prior", shouldn't that read "a priori"?*

Changed in several places.

- *P2, L24: I would rather write here "useful" instead of "promising"*

  Done.

- *P2, L44: move the reference in line 46 to line 44, otherwise these are a bit lost in line 46.*

  Done.

- *P9, Figure 4 bottom panel: In "Karlsruhe" the "l" is missing.*

  Thanks for noting this typo which has been corrected.

- *P10, L155: In the Figure 4 caption the average value is given as 1.1527*

  Thank you for noting this inconsistency due to a typo. The correct value is 1.1527, which has been corrected in the text.

- *P10, L166: two full stops, thus one obsolete.*

  Corrected.

- *P10, L166: List with comma: "of cloud water, total water amounts and isotope ratios."*

  Changed.

- *P10, L167: Deployment -> campaign or field mission*

  Changed to "field mission".

- *P10, L127: 27 Sep 2018 and 21 Oct 2018 -> 27 September and 21 October 2018*

  Changed.

- *P11, L186: "above" obsolete*

  Deleted.

- *P11, L188: in loc. cit.????*

  That is short for "loco citato", Latin for "in the place cited" (`https://en.wikipedia.org/wiki/Loc._cit.`). Instead of using this abbreviation, we have now repeated the citation.

- *P11, L192: Pearson coefficient -> Pearson correlation coefficient*

  Changed.

- *P11, L199: by small sensitivity -> by the small sensitivity*

  Changed.

- *P13, Figure 7, second panel: put -1 should be in superscript*

  Corrected.

- *P16, L254: ..procedure applied.. -> . . . procedure as applied. . . .*

  Changed.

- *P18, Figure 10 caption: Sep -> September*

  Changed.

- *P18, L281: Pearson coefficient -> Pearson correlation coefficient*

  Changed.

- *P23, L335: add "still", so that it reads . . . is still very sparse. . . . .*

  Added.

- *P23, L309: moistest -> moisture*

  Changed.

- *P23, L336: add also here "still" so that it reads "is also still sparse"*

  Added.

- *P25, Figure 15 caption: Jan -> January*

  Changed, also in Fig. 16 and 17.

**Review by Anonymous Referee #3**

*The manuscript by Schneider et al is a valuable contribution towards the application of global HDO/H2O data for scientific interpretation. While the paper in in general well written, I have a few points/concerns that require consideration.*

Thank you for your positive judgement of our contribution and for your review. In the following, all individual comments are quoted in italics and our response is given below.

**Major points:**

**Retrieval:**

1. *In table 1, you list a reduced chi2 filter of <150. Is this really the reduced chi2? If yes, 150 as a cutoff appears extremely high, i.e. you are not fitting the spectra properly. Given your regularizations, I would really like to see typical residuals of your spectral fits.*

   We have added a plot of a residual of a scene over the Sahara with high reduced $\chi^2$ (around 130). The fit is good and the residual looks normal. The very high signal at the bright scene amplifies errors due to inaccuracies in the spectroscopy and thus results in high reduced $\chi^2$. That could in principle be eliminated by a different normalization.

2. *Surface albedo prior: I am somewhat concerned about your choice of a surface albedo prior, using an annual average. This could create seasonal biases in your retrieval, depending on how strongly you regularize the retrieval. Some more details would be good to show that this is NOT the case. Also, what do you assume as the spectral dependence of your surface albedo in the fitting window? Varying snow cover and vegetation growth can cause large seasonal cycles in albedo, so using an annual average as prior seems to be not a great choice (MODIS provides more than enough data to get monthly priors).*

   We assume a linear spectral dependence of surface albedo in the retrieval window. The spectral slope is also derived in the inversion but not regularized. The regularization of the albedo at the centre of the spectral window is weak to give the algorithm enough freedom to adapt it to the actual situation. We have tested potential seasonal effects for scenes collocated to TCCON measurements by performing retrievals using a priori surface albedo from the (D)LER data product from the S5P+ Innovation Aerosol Optical Depth (AOD) and Bidirectional Reflectance Distribution Function (BRDF) project (Tilstra, 2021), which features monthly values. The results are very similar to those with the one-year average a priori, although the changed albedo prior results in slightly less convergences.

3. *Averaging kernels and profile scaling: It seems you are using a profile scaling approach, which might explain the somewhat counter-intuitive averaging kernels shown in Figure 10. As you only scale the profile: How do you compute (and provide) the column averaging kernels with your retrievals? It seems the data would be rather unusable without the kernels. Also, why did you choose not to fit the profile? While your DOF might not be ≫ 1, it would help not getting extreme values in your column kernel. In Figure 10, I would also suggest to plot the kernels with pressure as y axis. Given the scale height of H2O is low, the higher altitudes are rather unimportant for SWIR HDO/H2O retrievals.*

The data product provides column averaging kernels for each individual TROPOMI ground pixel. These are computed as described by Borsdorff et al. (2014) who showed that a total column averaging kernel can also be computed analytically for a profile scaling retrieval. We agree that providing the column average kernel to the user is important.

The data do not contain enough information to do a profile retrieval (DOF $\approx$ 1).

Averaging kernel plots are now shown with pressure as y axis.

**Science:**

*In your example cases, it would be good to really point out what could be learned from delta-D rather than just H2O alone. At the moment, this is unclear. More Rayleigh plots (e.g. a density plot of your global dataset) would be very helpful.*

Thank you for this important comment, in particular the idea of showing an additional ($H_2O$, $\delta D$) plot for the global dataset to highlight the additional use of $\delta D$. We have added a new figure depicting the ($H_2O$, $\delta D$) distributions for September 2018 over tropical lands and ocean and shortly discuss in Section 5.1 examples of what can be learned from the additional use of the $\delta D$ data from the scattering retrieval. In Section 5.2, we have pointed out the new insight from $\delta D$ by adding the following sentence at L391:

"Vertical mixing between the boundary layer and the free troposphere, such as during the moistening of the cold sector is one key process for which isotopes could provide additional information compared to total column $H_2O$ only."

However, it is not the role of this technical study to highlight the benefit of $\delta D$ for the study of different moist atmospheric processes, for this we refer to targetted scientific papers such as Risi et al. (2021), Thurnherr et al. (2021), Aemisegger et al. (2021) and many more.

**Small issues:**

- *Line 40: is notified?? I think I know what you mean but it won't be clear*

  This sentence is rephrased as follows: "Any loss of sensitivity to the partial column below the cloud is reflected in the column averaging kernel."

- *Line 61: Absorption cross section (not scattering)*

  Changed.

- *Line 80: Interferences and biases: Would be good to show spectral fits to maybe provide some more evidence to the gut feelings expressed here*

  We add a plot of a spectral fit.

- *Line 134: just saying "unit vector" is fine*

  Actually, it is not a unit vector $(1,0,0,\dots)$ but a vector with ones in each place $(1,1,1,\dots)$. We have clarified the sentence by rephrasing it to "a vector with ones in all places".

- *Figure 18: In the left panel, there seems to be a high density region with very low H2O crossing over all possible delta-D values. This seems somewhat unphysical, do you have an explanation for that? Could you plot the locations of this weird "vertical stripe" of data in the density plot?*

We have found a bug in the original plot script for the Rayleigh plot that resulted in the use of wrong filter parameters (thresholds). That has been corrected in the new version. However, there is still a stripe in the new version of the plot. The requested plot with the location of the low humidity data is shown in Figure 1. Currently it is unclear whether the stripe is an artefact. The issue will be further investigated.

[Figure]

Figure 1: TROPOMI single overpass results of XH$_2$O (**a**) and $\delta$D (**b**) for H$_2$O columns below $5 \times 10^{21}$ cm$^{-2}$ on 19 January 2020

**References**

Aemisegger, F., Vogel, R., Graf, P., Dahinden, F., Villiger, L., Jansen, F., Bony, S., Stevens, B., and Wernli, H.: How Rossby wave breaking modulates the water cycle in the North Atlantic trade wind region, Weather Clim. Dynam., 2, 281–309, https://doi.org/10.5194/wcd-2-281-2021, 2021.

Borsdorff, T., Hasekamp, O. P., Wassmann, A., and Landgraf, J.: Insights into Tikhonov regularization: application to trace gas column retrieval and the efficient calculation of total column averaging kernels, Atmos. Meas. Tech., 7, 523–535, https://doi.org/10.5194/amt-7-523-2014, 2014.

Risi, C., Muller, C., and Blossey, P.: Rain Evaporation, Snow Melt, and Entrainment at the Heart of Water Vapor Isotopic Variations in the Tropical Troposphere, According to Large-Eddy Simulations and a Two-Column Model, J. Adv. Model. Earth Syst., 13, e2020MS002 381, https://doi.org/10.1029/2020MS002381, 2021.

Scheepmaker, R. A., aan de Brugh, J., Hu, H., Borsdorff, T., Frankenberg, C., Risi, C., Hasekamp, O., Aben, I., and Landgraf, J.: HDO and H$_2$O total column retrievals from TROPOMI shortwave infrared measurements, Atmos. Meas. Tech., 9, 3921–3937, https://doi.org/10.5194/amt-9-3921-2016, 2016.

Schneider, A., Borsdorff, T., aan de Brugh, J., Aemisegger, F., Feist, D. G., Kivi, R., Hase, F., Schneider, M., and Landgraf, J.: First data set of H$_2$O/HDO columns from the Tropospheric Monitoring Instrument (TROPOMI), Atmos. Meas. Tech., 13, 85–100, https://doi.org/10.5194/amt-13-85-2020, 2020.

Thurnherr, I., Hartmuth, K., Jansing, L., Gehring, J., Boettcher, M., Gorodetskaya, I., Werner, M., Wernli, H., and Aemisegger, F.: The role of air–sea fluxes for the water vapour isotope signals in the cold and warm sectors of extratropical cyclones over the Southern Ocean, Weather Clim. Dynam., 2, 331–357, https://doi.org/10.5194/wcd-2-331-2021, 2021.

Tilstra, L.: TROPOMI ATBD of the directionally dependent surface Lambertian-equivalent reflectivity, KNMI report S5P-KNMI-L3-0301-RP, Royal Netherlands Meteorological Institute (KNMI), URL `https://d37onar3vnbj2y.cloudfront.net/static/surface/albedo/documents/s5p_dler_atbd_v1.1.0_2021-05-12_signed.pdf`, 2021.

Wunch, D., Toon, G. C., Sherlock, V., Deutscher, N. M., Liu, C., Feist, D. G., and Wennberg, P. O.: The Total Carbon Column Observing Network's GGG2014 Data Version, Tech. rep., TCCON, https://doi.org/10.14291/tccon.ggg2014.documentation.R0/1221662, 2015.

---

## Author Response (AR2)

**Review by Anonymous Referee #1**

The authors have considered all my comments and revised their manuscript accordingly. This is a very nice and valuable study and deserves to be published. I only have some minor technical issues that remain:

Thank you for your positive feedback as well as for reading thoroughly and spotting mistakes.

• P1, L4: Write (to be more clear) "the extension of the clear-sky to cloudy scenes"

We have rephrased the sentence as follows to be more clear: "Compared to the previous clear-sky-only data product, coverage is greatly enhanced by including scenes over low clouds, ..."

• P1, L10-11: I guess both holds for the mid and low latitudes. Thus, I would suggest to rewrite the sentence as follows: "In low and mid latitudes, the bias is small at low-altitude stations, but has a larger value at high altitude stations." Note, I exchanged here "latitude" by "altitude" since I guess you meant here "altitude" and not "latitude". There is another occasion in the manuscript where I think it was also written "latitude" instead of "altitude". Please check.

We do mean high latitude stations (at low altitudes). The whole sentence is about low altitude stations, which is the reason our original sentence starts with that phrase, and then distinguishes between low and mid latitudes on the one hand and high latitudes on the other hand (both at low altitudes). To improve understandability, we have rephrased it as follows: "At low-altitude stations, the bias is small at low and middle latitudes and has a larger value at high latitudes."

• P1, L14: Plays a role for what exactly? Please add.

We have rephrased this sentence as follows: "... since the information is filled up by the prior, a realistic shape of the prior is important for realistic total column estimation in these cases."

• P3, L74: Abbreviation TM5 not introduced.

We now write "the global chemistry Transport Model, version 5 (TM5)".

• P4, L92: Add "to" so that it reads "....and to stabilize....".

Done.

• P7, Fig 3 caption: Use capital letters, thus "RMS" instead of "rms".

Changed.

• P9, L139: A closing parentheses is missing in this sentence.

Added after "(a. s. l.)".

• P9, L140–142: This is still difficult to understand. On P17, L261 you give a much better explanation. I thus would suggest to write similar here as there.

We have rephrased it as follows: "If the altitude difference between station and satellite ground pixel is too large, both observe too different partial columns which leads to errors. That is the case for high-altitude stations that are typically located on mountains so that most co-located ground pixels have significantly lower surface height."

• P9, L147: Instead of "i is TROPOMI and j is TCCON" I would suggest to write "i denotes (or is used) for TROPOMI and ......"

We have changed "is" to "denotes".

• P10, L158: theses  $\mapsto$  these

Corrected.

• P10, L160: "are biased to each other" I think this is not correct English wording. I would suggest to rephrase this.

We have changed this to "do not agree".

• P10, L164: below? Where exactly? Please add this to the sentence.

We have changed "below" to "in the following" as this derivation is described in the remainder of that paragraph.

• P11, Fig 5 caption: (a) is not a histogram, but a correlation. Further, you should put the labels for the respective panels behind the text that describes what is shown, as it is done also for the other figures. Add also what the dashed line is showing.

To be precise, (a) shows a two-dimensional histogram of correlations. The dashed line represents the one-toone line. The figure caption is updated accordingly. In order to unify the placement of the labels (a), (b), etc., we have moved them to the front position in the other figures since we think that makes more clear which sentence belongs to which panel.

• P13, L218: Better to write "explained" instead of "connected"?

We have changed "connected to" to "explained by".

• *P14, Fig 17 caption: Also mention here in the caption what is shown by the colours and that the respective fits are shown.*

Added.

• P17, L240: Better to write "generally" instead of "typically"?

We have moved "typically" further behind in the sentence (now before "large variations"), since we mean that large variations in surface altitude and albedo within a ground pixel are typical for Garmisch, although some ground pixels may lie in the valley only and thus not have these large variations.

• *P17*, *L250: I would rather write "former" instead of "old", but if you have a version number you could also use that one here.*

We have changed "old" to "former" and also inserted "former" in the first sentence of the subsection.

• P21, L302: "....as for TCCON"  $\mapsto$  "....as derived for TCCON"

Changed.

• P21, L309: Add a short explanation why or refer to the respective section where you discuss this.

We have added a half-sentence explaining why and a reference to Section 4.1.

• P22, Fig 13 caption: Put (a), (b) behind the text describing the respective panels as it is done for the other figures.

To unify the placement of the labels of the panels (a), (b), etc., we have moved them to front position in the other figures.

• P23, Fig 14 caption: Add also here that fitted lines are shown.

Done similarly as in Fig. 7.

• P25, Fig 16 caption: Same as for Fig 13 caption.

See answer for Fig. 13.

• *P27, L353: State more clearly that you mean the data that has been presented in this manuscript.* We now write "these TROPOMI data".

**Review by Anonymous Referee #3**

The authors have, in a broad sense, responded well to most comments but have been deflecting at times. A few comments:

Thank you for your review.

• While showing the spectral fits, the authors state that "Nevertheless, the fit is reasonable". I don't know what that actually means. In fact, I consider the fit to be pretty mediocre and a reason for concern. The systematic residuals by far exceed the noise estimates and are systematic throughout the entire spectral range. I would be very interested if the authors actually find reduced chi2 that are anyhwere close to one in a typical case (i.e. not exceedingly dark). As the authors know, the entire posterior error analysis assumes no forward model error, which is clearly not the case. Thus, I expect systematic errors to far exceed your estimated precision errors. I would say this warrants a discussion (esp. for moving forward) instead of stating that they look "reasonable". E.g. in panel B, I see multiple lines that are saturated (H2O absorption) in the model but clearly not in the measurement. For some reason, I would have expected opposing behavior in residuals for some of the other H2O lines, as the fit would otherwise be able to adjust H2O to match the observation.

The radiance noise estimate is taken from the L1B files and is based on the read noise and shot noise as determined in the detector characterization, i. e. purely the statistical noise in the detector signal, but it does not include errors due to correction (offset, dark current, memory, PRNU, straylight) and conversion steps in the processor. Although these errors are mostly systematic, they should be taken into account partially. That means that the noise estimation does not include all components. Therefore, the use of this radiance noise estimation causes the residuals to exceed the noise estimate and leads to high  $\chi^2$  values. Thus,  $\chi^2$  does not correspond to the real fit quality. That happens especially for bright scenes (e. g. over the Sahara) where the sensor noise is very small so that the real noise is significantly underestimated. The spectral fit in our previous version shows the most extreme case with  $\chi^2$  very near to the filter threshold. Nevertheless, the absolute value of the residual is about 2% of the spectrum. To be more illustrative, we now show a typical case which is randomly chosen among clear-sky scenes co-located with Karlsruhe TCCON station near equinox. The reduced  $\chi^2$  value is 2.5.

• Figure 17: I honestly don't fully understand what I am seeing. While these are single measurements, as opposed to an average as in Figure 16, I would expect them to be broadly consistent in range and distribution. However, the dynamic range in H2O is minute while delta-D covers kind of everything, including (in my mind) somewhat unphysically low delta-D values for the tropics (especially at that high H2O concentration). It seems unlikely to get column delta-D values that are more depleted that the stratosphere. The discussion of this plot is also rather speculative to be honest.

Thank you for bringing up this problem. Actually, a bug in the plot script caused the contour lines to be shown transposed (i. e.  $H_2O$  and  $\delta D$  axes swapped). This has been fixed now. In this context, the description of the plot has been changed as well. We agree that the description is to some degree speculative. Thus, we have changed "is most likely" to "may be".

• Figure 12: I would have preferred a linear pressure scale. The scale height of H2O is really low, so this figure emphasized pressure (i.e. height) ranges that are somewhat irrelevant. A linear pressure scale would emphasize the lower height ranges (say 0–3km) more.

The averaging kernel plots now have a linear pressure scale.

• "That has been corrected in the new version. However, there is still a stripe in the new version of the plot. The requested plot with the location of the low humidity data is shown in Figure 1. Currently it is unclear whether the stripe is an artefact. The issue will be further investigated."

Have you looked, at least in a cursory fashion, at a potential cause for this artifact? It seems important to me to figure that out, rather that just deferring to further analysis. It can't take too much time to at least dig into it a bit deeper than the authors did so far. It would be a shame to have a plot with an obvious problem in a peer reviewed publication, as a reviewer I shouldn't really allow it. A simple explanation might be that the error in delta-D blows up if H2O is low (but this can be filtered).

Before our last answer, we have looked at several quantities such as column averaging kernels at various altitudes near the surface, surface albedos, effective cloud parameters, reduced  $\chi^2$  of pre-fit and final. We show most of these plots in Fig. 1 and 2. We do not see any significant differences to other regions where  $\delta D$  values are considered realistic. For example, the ground pixels you consider artefacts have increased averaging kernel values as expected below clouds, but other ground pixels with realistic  $\delta D$  values have similar averaging kernel values.

With a filter for low H2O the stripe could be eliminated by construction, however filter criteria should not be based on a single feature in a single overpass in a single orbit. It is important to consider effects for different types of scenes when changing filter parameters. Filtering out H2O columns below  $7 \times 10^{21}$  molec cm-2 (which removes the stripe in question) slightly worsens the validation, although the difference is small. This does not support such a filter for the absolute value of H2O.

The stripe consists of a relatively low number of ground pixels that were highlighted due to the logarithmic colour scale of the plot. We have changed the colour scale to linear to give a better impression.

**Editor comments**

Thank you for your suggestions for improvement.

**Technical corrections:**

• Abstract: A key feature of the retrieval is that also measurements under cloudy conditions are considered but only if the cloud height is low. However, the abstract reads as if all clouded observations are considered. Please specify this important selection of low clouds only already in the abstract.

This is now mentioned in the third sentence of the abstract.

• Lines 35/36: Consider reformulation; in the current text, the reader might refer the "upgrade" of the spatial resolution to Sentinel 5.

We now write " $5.5 \text{ km} \times 7 \text{ km} (7 \text{ km} \times 7 \text{ km} \text{ before August 2019})$ ".

• Line 62: Please provide an argument why H218O can be fixed in the retrieval.

We have added the half-sentence "since the absorption is very weak".

• TCCON: I propose to also add a map of all stations, or to add the stations to Fig. 16 (a).

Locations of the stations shown in Fig. 9 have been added to Fig. 16a.

• Fig. 18: Colormaps are somehow a matter of taste, but since clouds are white I was first confused that the cloud map uses white for indicating no clouds. You might consider a different colormap e.g. from darkblue or dark gray to white.

The colormap for cloud fraction has been changed to blues (from blue to white).

• The author contributions are quite vague. It is not clear who did all the analysis. Please specify.

We now specify the contributions as follows: "AS made the retrievals and performed the analysis with help from TB, JadB, AL and JL."